# Contextual Optimization Under Model Misspecification: A Tractable and Generalizable Approach

**Omar Bennouna** [* 1]    **Jiawei Zhang** [* 1 2]    **Saurabh Amin** [1]    **Asuman Ozdaglar** [1]

## Abstract

Contextual optimization problems are prevalent in decision-making applications where historical data and contextual features are used to learn predictive models that inform optimal actions. However, practical applications often suffer from model misspecification due to incomplete knowledge of the underlying data-generating process, leading to suboptimal decisions. Existing approaches primarily address the well-specified case, leaving a critical gap in handling misspecified models. In this paper, we propose a novel Integrated Learning and Optimization (ILO) framework that explicitly accounts for model misspecification by introducing a tractable surrogate loss function with strong theoretical guarantees on generalizability, tractability, and optimality. Our surrogate loss aligns with the true decision performance objective, ensuring robustness to misspecification without imposing restrictive assumptions. The proposed approach effectively mitigates the challenges of non-convexity and non-smoothness in the target loss function, leading to efficient optimization procedures. We provide rigorous theoretical analysis and experimental validation, demonstrating superior performance compared to state-of-the-art methods. Our work offers a principled solution to the practically relevant challenge of model misspecification in contextual optimization.

## 1. Introduction

Many real-world decision-making problems require optimizing decisions based on uncertain parameters that can be estimated using contextual information. A canonical example is traffic routing, where the costs of traversing network edges—initially unknown—depend on observable features such as weather, time of day, and construction activity. Given historical data, the goal of contextual optimization is to learn a policy that maps observed contexts to near-optimal decisions (Sadana et al., 2024).

A standard framework for contextual optimization assumes that the unknown cost vector $c \in \mathbb{R}^d$ is correlated with an observed context $x \in \mathbb{R}^k$, and the objective is to minimize expected decision cost:

$$\min_{\pi \in \Pi} \mathbb{E}_{(x,c) \sim P} \left( c^\top \pi(x) \right). \qquad (1)$$

Here, $P$ is the joint distribution of $(x, c)$, and $\Pi$ is a set of feasible policies that map contexts to decisions in a bounded convex set $W \subset \mathbb{R}^d$. Since the true cost function is unknown in practice, it must be approximated using a prediction model from a hypothesis class $\mathcal{H}$.

The widely used predict-then-optimize approach first learns a predictor $\hat{c}(x)$ from $\mathcal{H}$ and then selects a decision by solving:

$$\hat{\pi}(x) \in \arg \min_{w \in W} \hat{c}(x)^\top w. \qquad (2)$$

Prior works mainly adopt two approaches: *Sequential Learning and Optimization (SLO)* (Bertsimas and Kallus, 2020; Hu et al., 2022) and *Integrated Learning and Optimization (ILO)* (Donti et al., 2017; Elmachtoub and Grigas, 2022; Sun et al., 2023). SLO focuses solely on minimizing prediction error for $c$, ignoring the downstream optimization task, while ILO prioritizes decision performance over accurate cost prediction. Recent research in Integrated Learning and Optimization (ILO) refines cost prediction to improve downstream decisions, under the assumption that $\mathcal{H}$ contains the true cost function. In other words, previous ILO approaches assume that $\mathcal{H}$ is well-specified.

However, real-world settings often violate the above assumption. We say that $\mathcal{H}$ is misspecified when it does not contain a predictor that perfectly captures the true cost function. In real-world applications, this situation can arise due to incomplete feature sets, unmodeled dependencies, or distribution shifts. Despite its practical significance, contextual

---

[*]Equal contribution  [1]Department of EECS, Massachusetts Institute of Technology [2]Department of Computer Sciences, University of Wisconsin–Madison. Correspondence to: Omar Bennouna <omarben@mit.edu>, Jiawei Zhang <jzhang2924@wisc.edu>.

*Proceedings of the 42^{nd} International Conference on Machine Learning*, Vancouver, Canada. PMLR 267, 2025. Copyright 2025 by the author(s).

optimization under model misspecification remains under-explored in the literature. Most existing research in ILO assumes $\mathcal{H}$ is well-specified and focuses on design of surrogate loss functions and algorithms to maximally reduce the decision error. While some recent empirical works (Donti et al., 2017; McKenzie et al., 2023; Kotary et al., 2023; Huang and Gupta, 2024) explore decision-aware learning under misspecification, they lack theoretical guarantees for global optimality and generalization. Additional literature review is available in Appendix A.3.

We argue that the key challenge lies in distinguishing between two forms of misspecification (see Appendix A.4 for further details):

1. **Prediction Misspecification:** The inability of a model $\hat{c}(x)$ to accurately estimate $c(x)$;

2. **Decision Misspecification:** The inability of a model $\hat{c}(x)$ to produce the same decisions as those derived from the true cost function.

Thus, in contextual optimization, the challenge is to design tractable approaches to tackle decision misspecification. Indeed, a model can yield poor predictions yet still make optimal decisions–and vice versa. Yet, standard methods such as Sequential Learning and Optimization (SLO) and Smart Predict-then-Optimize (SPO+) (Elmachtoub and Grigas, 2022) fail to guarantee optimal decision-making in the presence of misspecification, leading to suboptimal policies.

### 1.1. Problem setup

For simplicity, we consider that unknown parameters appear linearly in the objective function of the stochastic optimization problem:

$$\min_{w \in W} \mathbb{E}_{(x,c) \sim P} \left( c^\top w | x \right) = \min_{w \in W} \mathbb{E}_{(x,c) \sim P} (c|x)^\top w. \quad (3)$$

Since $c$ is unknown, we approximate it using a predictor $\hat{c} \in \mathcal{H}$, where $\hat{c}(x)$ is learned using historical data. Importantly, $\mathcal{H}$ can be misspecified. The decision-making process is as follows: first, observe the context $x$ and predict the cost vector $\hat{c}(x)$. Next, compute a decision $w(\hat{c}(x))$ that minimizes the predicted cost:

$$w(\hat{c}(x)) \in \arg\min_{w \in W} \hat{c}(x)^\top w. \quad (4)$$

We consider that the hypothesis set $\mathcal{H}$ is parametric, i.e. it can be written as $\mathcal{H} = \{\hat{c}_\theta, \ \theta \in \mathbb{R}^m\}$. Our goal is to find a predictor $\hat{c}_\theta$ that minimizes decision error, defined as:

$$\ell_P(\theta) = \mathbb{E}_{(x,c) \sim P} \left( \max_{w \in W^\star(\hat{c}_\theta(x))} c^\top w \right), \quad (5)$$

where $W^*(\tilde{c}) := \arg\min_{w \in W} \tilde{c}^\top w$ represents the set of optimal solutions given a predicted cost vector $\tilde{c}$.

To illustrate why traditional approaches such as SLO and (SPO+) fail to handle misspecified $\mathcal{H}$, consider a simple binary classification example: suppose the decision space is $W = [-1/2, 1/2]$, and the cost vector $c$ takes values in $\{-1, 1\}$, and the minimization in (3) is replaced by maximization. The optimization problem then reduces to minimizing:

$$\ell_P(\hat{c}) = -\frac{1}{2} + \mathbb{E}_{(x,c) \sim P} \left( \mathbb{1}_{\text{sign}(\hat{c}(x)) \neq c} \right). \quad (6)$$

This formulation is closely related to minimizing classification error in binary classification. We can observe that SLO and SPO+ lead to suboptimal solutions in such a setting.

**Example 1.** *Consider the following hypothesis class: $\mathcal{H} = \{\hat{c}_1, \hat{c}_2\}$, and the context $x$ takes only two values 1 and 2 equally likely, and $c$ always takes the value 1. Assume that $\hat{c}_1(1) = \frac{1}{8}$, $\hat{c}_1(2) = \frac{1}{8}$, $\hat{c}_2(1) = 1$, and $\hat{c}_2(2) = -\frac{1}{6}$. Although $\hat{c}_1$ leads to optimal decision-making, both SPO+ and SLO select $\hat{c}_2$, a suboptimal predictor (refer to Appendix A.1).* ◁

This example demonstrates that SPO+ and SLO do not necessarily yield optimal decisions. Similar conclusions can be drawn for a more practical traffic routing problem, where the goal is to allocate traffic across roads to minimize travel time based on contextual factors like weather, time of day, and road construction. Suppose the true travel cost depends on all three factors, but due to model limitations, only weather and time of day are considered, ignoring road construction effects. This model misspecification leads both SLO and SPO+ to learn systematically biased cost estimates, which fail to anticipate congestion caused by construction. As a result, traffic is misallocated, leading to longer travel times during construction periods. SLO fails because it prioritizes prediction accuracy over decision performance, while SPO+ fails because its decision-aware learning is constrained by an incomplete hypothesis class. This demonstrates how standard methods can yield suboptimal decisions, even if predictions appear reasonable, highlighting the need for a method that directly optimizes decision quality despite misspecification.

### 1.2. Our contribution

To address these limitations, we propose a novel decision-aware surrogate loss function that explicitly minimizes decision error regardless of whether or not $\mathcal{H}$ is misspecified. Our approach ensures optimal decision performance under model misspecification and provides strong theoretical guarantees, ensuring both optimality and generalization performance. Importantly, our tractable surrogate loss function matches the optimal solutions of the true objective (5), has favorable optimization properties, avoiding bad local minima, and generalizes well to new distributions. We prove

that our surrogate loss:

- Yields globally optimal decision policies (**Theorem 1**).

- Exhibits strong generalization properties (**Theorem 2**).

- Is computationally tractable, enabling efficient optimization via first-order methods (**Theorem 4**).

We also evaluate our approach against SPO+ and SLO, showing superior performance under varying degrees of misspecification.

The remainder of the paper is structured as follows. Section 2 formalizes the problem and introduces our surrogate loss function. Section 3 establishes theoretical guarantees. Section 4 presents empirical results comparing our approach to existing methods. Section 5 concludes with future research directions.

## 2. Our approach

### 2.1. Rewriting $\ell_P$ under a generic assumption

Recall from (5) that the generic form of the target loss function in our contextual optimization problem involves the worst case value of $c^\top w$ when $w \in \arg\min_{w \in W} \hat{c}_\theta(x)^\top w$ because the solution to $\min_{w \in W} \hat{c}_\theta(x)^\top w$ is not necessarily unique in theory. In practice, however, $\arg\min_{w \in W} \hat{c}_\theta(x)^\top w$ is likely to be unique. For example, in the common case where $\hat{c}_\theta(x)$ is a continuous random variable and $W$ is a polyhedron (or more generally when the set of directions $\tilde{c}$ for which the solution to $\min_{w \in W} \tilde{c}^\top w$ is not unique is of Lebesgue measure equal to zero), uniqueness holds with probability 1. Hence, due to practical reasons and theoretical convenience, we make the following assumption henceforth:

**Assumption 1.** *For any $\theta \in \mathbb{R}^m$ such that $\hat{c}_\theta(x) \neq 0$ almost surely, $\min_{w \in W} \hat{c}_\theta(x)^\top w$ has a unique solution with probability 1 when $(x, c) \sim P$.*

We denote $W_P$ the set of measurable mappings from the support of the joint probability $P$ of $(x, c)$ to the set of feasible decisions $W$. Under Assumption 1, we can see that the problem of minimizing target loss can be written as follows:

$$\min_{\theta \in \mathbb{R}^m} \ell_P(\theta) = \min_{\theta \in \mathbb{R}^m} \min_{w_P \in W_P} \mathbb{E}_{(x,c) \sim P} \left( c^\top w_P(x) \right) \quad (7)$$

$$\text{s.t. } \forall x \in \mathbb{R}^k, \ w_P(x) \in \arg\min_{w \in W} \hat{c}_\theta(x)^\top w. \quad (8)$$

We formulate a proposition that provides a sufficient condition for Assumption 1 to hold when $W$ is a polyhedron in Appendix A.5.

Our task now becomes to solve the optimization problem (7). Since known approaches cannot be directly applied to solve (7), we develop an approach based on a surrogate loss function that enjoys two properties:

1. **Consistency:** The optimal solutions for the surrogate loss are also optimal for the target loss;

2. **Tractability:** The surrogate loss is tractable to optimize.

### 2.2. Introducing our surrogate loss function

We introduce a new surrogate loss function – referred as Consistent Integrated Learning and Optimization (CILO) loss – for contextual optimization under misspecification.

**Definition 1.** *(CILO loss) For $\beta \in \mathbb{R}$, we define the function $\ell_P^\beta$ as*

$$\forall \theta \in \mathbb{R}^m, \ \ell_P^\beta(\theta) := \min_{\overline{w}_P^\beta \in \overline{W}_P^\beta} \mathbb{E}_{(x,c) \sim P} \left( \hat{c}_\theta(x)^\top \overline{w}_P^\beta(x) \right)$$

$$- \min_{w_P \in W_P} \mathbb{E}_{(x,c) \sim P} \left( \hat{c}_\theta(x)^\top w_P(x) \right),$$

*where $\overline{W}_P^\beta = \{ \overline{w}_P \in W_P, \ \mathbb{E}_{(x,c) \sim P} \left( c^\top \overline{w}_P(x) \right) \leq \beta \}$.*

In (Elmachtoub and Grigas, 2022), authors show that in the binary classification setting, the SPO+ loss is equal to the hinge loss, i.e.

$$\forall c, \tilde{c} \in \mathbb{R}^d, \ \ell_{\text{SPO+}}(\tilde{c}, c) = \max(0, 1 - 2c\tilde{c}).$$

We introduce the following notation, for every $\beta \in \mathbb{R}$ and measurable function $\hat{c}$,

$$\ell_P^\beta(\hat{c}) = \min_{\overline{w}_P^\beta \in \overline{W}_P^\beta} \mathbb{E}_{(x,c) \sim P} \left( \hat{c}(x)^\top \overline{w}_P^\beta(x) \right)$$

$$- \min_{w_P \in W_P} \mathbb{E}_{(x,c) \sim P} \left( \hat{c}(x)^\top w_P(x) \right).$$

We denote $\beta_{\mathcal{H},P}^\star := \min_{\hat{c} \in \mathcal{H}} \ell_P(\hat{c})$ and $\beta_{\max,P} = \mathbb{E}_{(x,c) \sim P}(\max_{w \in W} c^\top w)$. The following proposition shows that the CILO loss also has an interesting expression in the binary classification setting when the hypothesis set is decision-well-specified.

**Proposition 2.1.** *In the binary classification setting, when $\beta = \beta_{\mathcal{H},P}^\star$ and $c$ is a deterministic function of $x$, for every measurable function $\hat{c}$, when $\mathcal{H}$ is decision-well-specified, we have*

$$\ell_P^\beta(\hat{c}) = \mathbb{E}_{(x,c) \sim P}(|\hat{c}(x)| \, \mathbb{1}_{sign(\hat{c}(x)) \neq c}). \quad (9)$$

The proof of this proposition is in Appendix A.6. We can see in Proposition 2.1 that the CILO loss nearly matches the expression of the target loss when the hypothesis set is

decision-well-specified (and not necessarily prediction-well-specified) and $c$ is deterministic in $x$. Furthermore, when $\hat{c}$ does not take the value zero, the consistency or the CILO loss is clear from its expression.

The following example shows that whereas SLO and SPO+ fail to provide a minimizer for $\ell_P$, this new surrogate successfully does so.

**Example 2.** *We revisit Example 1. We have seen that SPO+ and SLO do not favor the best element of the hypothesis set $\mathcal{H}$. Since $\mathcal{H}$ is decision-well-specified, by choosing $\beta = \beta^\star_{\mathcal{H},P} = -\frac{1}{2}$, from proposition 2.1, equality (9) is satisfied. Consequently, we have $\ell^\beta_P(\hat{c}_1) = 0$ and $\ell^\beta_P(\hat{c}_2) = \frac{1}{12} > \ell^\beta_P(\hat{c}_1)$. We see that indeed here, the global minimum of this new loss is optimal for the target loss.* ◁

It is important to keep in mind that in the example above, the model is prediction misspecified, but decision well-specified. Hence, even when there exists a cost predictor that yields the same decisions as $c$, SPO+ and SLO may fail to favor the best cost predictor in term of decisions. Decision-well-specification is not necessary for the CILO loss' consistency to hold in the example. To see that, it suffices to add a third point in the distribution of $P$ which is labeled the same way by $\hat{c}_1$ and $\hat{c}_2$. This example will yield the same results.

We will now prove that the consistency observed in example 2 holds in general. The lemma below will be useful when proving our main consistency theorem.

**Lemma 1.** *For every $\beta \in \mathbb{R}$ such that $\beta^\star_{\mathcal{H},P} \leq \beta$, $\ell^\beta_P$ is non-negative.*

We now formulate our main consistency theorem.

**Theorem 1.** *Let $\beta \in \mathbb{R}$ such that $\beta^\star_{\mathcal{H},P} \leq \beta < \beta_{max,P}$. Under Assumption 1, for every $\theta \in \mathbb{R}^m$ with $\hat{c}_\theta(x) \neq 0$ almost surely, $\ell_P(\theta) \leq \beta$ if and only if $\theta$ is a minimizer of $\ell^\beta_P$. In particular, when $\beta = \beta^\star_{\mathcal{H},P}$, $\theta$ is a minimizer of $\ell^\beta_P$ if and only if it is a minimizer of $\ell_P$.*

The proof of this theorem is in Appendix A.7.

This result suggests a natural approach to minimizing $\ell_P$: first, find a tractable method to minimize the CILO loss $\ell^\beta_P$ for a suitable choice of auxiliary parameter $\beta$, ensuring the candidate minimizer $\theta$ satisfies $\hat{c}_\theta(x) \neq 0$ almost surely. While $\beta^\star_{\mathcal{H},P}$ is unknown, achieving a sufficiently small value of $\ell^\beta_P$ during optimization likely ensures that the target loss $\ell_P$ remains below $\beta$. To select $\beta$, one can perform a line search over a suitable interval $[\underline{\beta}, \overline{\beta}]$ and choose the value that minimizes the CILO loss. We now develop a tractable (in the sense that it runs in polynomial time in theory and is efficient in practice) procedure to optimize $\ell^\beta_P$—making the line search feasible—and formalize the relationship between near-optimal $\ell^\beta_P$ values and the target loss $\ell_P$. This addresses a key limitation in contextual optimization litera-

ture: existing surrogate losses, though tractable, guarantee optimality only under well-specified settings. While some perform well experimentally in misspecified cases, no existing approach ensures global optimality across all levels of misspecification. We close this gap in Section 3.

## 3. Technical approach

In this section, we formalize our technical approach based on the CILO loss function and the consistency result. We first mention three key issues that our approach seeks to address. In the remainder of the paper, we denote by $\|.\|$ the $L^2$ norm.

Firstly, we do not have access to the joint distribution P of random variables $(x, c)$; instead we assume that we have access to a historical dataset $S = \{(x_1, c_1), \ldots, (x_n, c_n)\}$, where $n$ is the number of samples and each sample $(x_i, c_i)$, $i \in [n]$ is sampled from $P$. Thus, we seek to optimize the empirical version of CILO, denoted $\ell^\beta_{P_n}$, where $P_n$ is the uniform distribution over the dataset $S$. The question we need to address is whether by minimizing empirical CILO, we can obtain good generalization performance (i.e., guarantees on out-of-sample CILO loss). Theorem 2 shows that this is indeed the case.

Secondly, the CILO loss $\ell^\beta_P$ is a non-convex and non-smooth function, and we need a technique to optimize it in a tractable manner. We provide one way to address this issue by leveraging the Moreau envelope smoothing technique (Sun and Sun, 2021) to transform the problem of minimizing $\ell^\beta_P$ into another optimization problem whose objective function enjoys good landscape; in particular it is smooth and has no "bad" first-order stationary points or local minima (see Theorem 4). Consequently, this smoothed optimization problem is conducive to gradient descent.

Thirdly, to guarantee consistency, we must ensure that the optimization procedure is able to find minimizer $\theta$ that verifies $\hat{c}_\theta(x) \neq 0$ almost surely (see Theorem 1). We address this issue by refining our smoothing procedure so that gradient descent on smoothed CILO loss results in a minimizer that verifies $\hat{c}_\theta(x) \neq 0$ almost surely (see Theorem 6).

These three steps together ensure that we have a tractable approach to minimizing empirical CILO loss, resulting in a solution that has good generalization perform optimality guarantee.

### 3.1. Generalization performance

We study the generalization performance of the empirical version of CILO loss and show that by optimizing the empirical CILO loss $\ell^\beta_{P_n}$, we can ensure a nearly optimal value for its out of sample counterpart $\ell^\beta_P$. We first make the following boundedness assumptions.

**Assumption 2.** *$W$ is closed and bounded and there exists $K \geq 0$ such that for all $x \in \mathbb{R}^k$, $\|c(x)\| \leq K$. We denote $B_W = \sup_{w \in W} \|w\|$.*

**Assumption 3.** *When $\theta = 0$, $\hat{c}_\theta(x) = 0$. Furthermore, there exists $B_\Phi \geq 0$ such that the gradient with respect to $\theta$, $\nabla \hat{c}_\theta(x)$, is piecewise continuous and bounded by $B_\Phi$ for every $\theta \in \mathbb{R}^m$ and $x \in \mathbb{R}^k$.*

The first part of Assumption 3 is natural when we assume that there exists $\theta_0 \in \mathbb{R}^m$ such that $\hat{c}_{\theta_0} = 0$. By reparametrizing the hypothesis set as $\mathcal{H}' = \hat{c}_{\theta + \theta_0}, \theta \in \mathbb{R}^m$, we obtain $\hat{c}_0 = 0$. Furthermore, a direct consequence of this assumption is that for every $x \in \mathbb{R}^k$ and $\theta \in \mathbb{R}^m$,

$$\|\hat{c}_\theta(x)\| = \left\| \int_0^1 \nabla \hat{c}_{t\theta}(x)^\top \theta \, dt \right\|$$
$$\leq \int_0^1 \left\| \nabla \hat{c}_{t\theta}(x)^\top \theta \right\| dt \leq B_\Phi \|\theta\|.$$

We now present our main generalization result.

**Theorem 2.** *Let $\beta \geq \beta_{\mathcal{H},P}^\star$. Let $\theta^\star \in \mathbb{R}^m$ such that $\ell_{P_n}^\beta(\theta^\star) \leq \varepsilon$. Assume that there exists $D \geq 0$ such that $\|\theta^\star\| \leq D$. Under assumptions 2, and 3, and assuming $\beta_{\mathcal{H},P}^\star > \beta_{min,P}$, with probability at least $1 - \delta$, we have*

$$\ell_P^\beta(\theta^\star) \leq \varepsilon + O\left( \frac{1}{\beta - \beta_{min,P}} \sqrt{\frac{\log \frac{1}{\delta}}{n}} \right).$$

Notice here that when the hypothesis set $\mathcal{H}$ is decision-misspecified, we have $\beta_{\mathcal{H},P}^\star > \beta_{\min,P}$, and consequently the term $\frac{1}{\beta - \beta_{\min,P}}$ is bounded. If $\mathcal{H}$ is (nearly) well-specified, we can choose a larger $\beta$ to guarantee the generalization bound to be good. This theorem implies that our surrogate loss can generalize when $n$ is large and hence minimizing the empirical version of the surrogate loss $\ell_{P_n}^\beta$ yields small optimality gap for its out-of-sample counterpart $\ell_P^\beta$. Using Theorem 1 and Theorem 2, we can deduce that finding $\theta$ such that $\hat{c}_\theta(x) \neq 0$ almost surely and minimizing $\ell_{P_n}^\beta$ yields an optimal value for $\ell_P(\theta)$.

Theorem 1 ensures that the closer $\theta$ is to optimality for $\ell_P^\beta$ for a well-chosen $\beta \in \mathbb{R}$, the closer it is to optimality for $\ell_P$. When $\theta$ is nearly optimal for $\ell_P^\beta$, we aim to quantify how close it is to optimality for $\ell_P$. By adopting a stability assumption regarding the linear optimization problem, we can establish a more concrete accuracy bound for the solution obtained from the empirical version $\ell_{P_n}^\beta$.

### 3.2. Stronger consistency result

For ease of presentation, we make the following assumption from now on:

**Assumption 4.** *For $\theta \neq 0$, $\hat{c}_\theta(x) \neq 0$ almost surely.*

Assumption 4 holds when for every $\theta \in \mathbb{R}^m \setminus \{0\}$, $\hat{c}_\theta$ is a nonzero analytic function and $x$ has a continuous distribution. In this case, the set of zeroes of $\hat{c}_\theta$ is of measure equal to zero (see (Mityagin, 2015)) and consequently $x$ does not belong to this set almost surely. This assumption, coupled with Assumption 1, implies that for any $\theta \in \mathbb{R}^m \setminus \{0\}$, the problem $\min_{w \in W} \hat{c}_\theta(x)^\top w$ has a unique solution almost surely.

We give a more concrete optimality guarantee obtained by minimizing $\ell_P^\beta$, which sharpens Theorem 1. We first make an assumption similar to the assumption in (Hu et al., 2022) which enables the authors to get good generalization guarantees for SLO in the well-specified case. However, our assumption add conditions on the hypothesis set rather than the ground truth cost.

**Assumption 5.** *Assume that $W$ is a polyhedron. We denote $W^\angle$ the set of extreme points of $W$. For a given context $x \in \mathbb{R}^k$, we denote for every $\theta \in \mathbb{R}^m$,*

$$\Delta_\theta(x) = \begin{cases} \displaystyle\min_{w \in W^\angle \setminus W^\star(\hat{c}_\theta(x))} \hat{c}_\theta(x)^\top w - \min_{w \in W^\angle} \hat{c}_\theta(x)^\top w \\ \qquad\qquad\qquad\qquad if\ W^\star(\hat{c}_\theta(x)) \neq W \\ 0 \qquad\qquad\qquad\qquad else. \end{cases}$$

*Assume that for some $\alpha > 0$ and $\gamma \geq 0$, for every $\theta \in \mathbb{R}^m$*

$$\forall t > 0, \quad \mathbb{P}(0 < \Delta_\theta(x) \leq \|\theta\| \, t) \leq \left( \frac{\gamma t}{B_W} \right)^\alpha.$$

The assumption above ensures the stability of the target loss $\ell_P$. Specifically, it strengthens the uniqueness guarantee provided by Assumption 1. Besides ensuring that $\min_{w \in W} \hat{c}_\theta(x)^\top w$ has a unique solution almost surely, it offers a sharper measure for how sensitive the mapping $w \longmapsto \hat{c}_\theta(x)^\top w$ is to deviations from $w(\hat{c}_\theta(x))$ with high probability. Assumption 5 is reasonable when $\hat{c}_\theta(x)$ has a continuous distribution, since it is equivalent to say that $\hat{c}_\theta(x)$ is likely to have a direction that is not too close to being perpendicular to one of the faces of the polyhedron $W$–that is, the probability of this direction to be falling within one of the red cones shown in Figures 1 and 4 decays to 0 as the cones get more narrow. More details about this assumption are provided in Appendix A.8.

The following theorem provides a relationship between optimality gaps of $\ell_P^\beta(\theta)$ and $\ell_P(\theta)$ when $\theta$ is bounded away from 0, i.e. there exists $u > 0$ such that $\|\theta\| \geq u$.

**Theorem 3.** *Under assumptions 2, 3 and 5, there exists $\alpha > 0$ such that for all $\theta \in \mathbb{R}^m$ bounded away from 0 and $\beta \geq \beta_{\mathcal{H},P}^\star$,*

$$\ell_P(\theta) \leq \beta + O\left( \ell_P^\beta(\theta)^{1 - \frac{1}{1+\alpha}} \right).$$

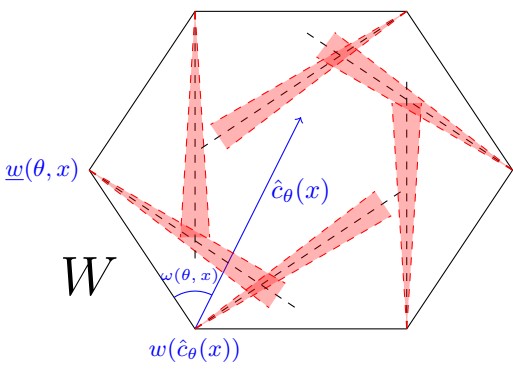

*Figure 1.* Less likely directions for $\hat{c}_\theta(x)$ relative to $W$ when it is a polyhedron in $\mathbb{R}^2$

The proof of this theorem can be found in Appendix A.12. Assuming that $\theta$ is bounded away from 0 is crucial here, because when $\theta = 0$, the uniqueness property in Assumption 1 is not guaranteed, and the consistency in Theorem 1 is not satisfied. Similarly, when $\theta$ is close to 0, $\ell_P^\beta$ exhibits poor behavior, as it approaches the scenario where consistency breaks down. Although Assumption 5 requires $W$ to be a polyhedron, it is possible to consider a similar yet reasonable assumption that does not impose this requirement. This alternative leads to a stronger consistency inequality (see Appendix A.8).

**Remark 1.** *This theorem suggests that to minimize $\ell_P$, one should seek an approximate minimizer $\theta$ of $\ell_P^\beta$ for a well-chosen $\beta \in \mathbb{R}$ with $\|\theta\|$ kept away from zero. By applying Theorem 2, it follows that it suffices to find some $\theta \in \mathbb{R}^m$ such that $\ell_{P_n}^\beta(\theta)$ is small while ensuring $\|\theta\|$ remains bounded away from zero.*

### 3.3. Optimizing our surrogate

Based on theorems 1, 2, and 3, it is sufficient to find a $\theta \in \mathbb{R}^m$ such that $\ell_{P_n}^\beta$ is small and $\theta$ is kept bounded away from zero to minimize $\ell_P$. The empirical surrogate loss $\ell_{P_n}^\beta$ can be written as:

$$\forall \theta \in \mathbb{R}^m, \; \ell_{P_n}^\beta(\theta) = g_{P_n}(\theta) - \overline{g}_{P_n}^\beta(\theta),$$

where

$$g_{P_n}(\theta) := -\mathbb{E}_{(c,x)\sim P_n}\left(\min_{w\in W}\hat{c}_\theta(x)^\top w\right) \quad (10)$$

$$= -\min_{w_{P_n}\in W_{P_n}}\mathbb{E}_{(c,x)\sim P_n}\left(\hat{c}_\theta(x)^\top w_{P_n}(x)\right) \quad (11)$$

$$\overline{g}_{P_n}^\beta(\theta) := -\min_{w_{P_n}\in \overline{W}_{P_n}^\beta}\mathbb{E}_{(c,x)\sim P_n}\left(\hat{c}_\theta(x)^\top \overline{w}_{P_n}^\beta(x)\right). \quad (12)$$

The equality in (11) is satisfied because choosing a policy $w_P$ minimizing $E_{(c,x)\sim P_n}\left(\hat{c}_\theta(x)^\top \overline{w}_{P_n}^\beta(x)\right)$ is equivalent to choosing $w_P(x)$ minimizing $\hat{c}_\theta(x)^\top w_P(x)$ for every $x$ in $\mathbb{R}^k$.

**Landscape properties and smoothing.** We take a closer look at the structure of our surrogate. We make the following mild assumption.

**Assumption 6.** *The function $\theta \longmapsto \hat{c}_\theta(x)$ is differentiable for every $x \in \mathbb{R}^k$, and there exists $B_L > 0$ such that for all $\theta, \theta' \in \mathbb{R}^m$ and $x \in \mathbb{R}^k$,*

$$\|\nabla_\theta \hat{c}_\theta(x) - \nabla_\theta \hat{c}_{\theta'}(x)\| \le B_L \|\theta - \theta'\|.$$

This assumption is sufficient for $\ell_{P_n}^\beta$ to be written as a difference of two convex functions. Specifically, we have the following result:

**Proposition 1.** *Under assumptions 2 and 6, $\ell_{P_n}^\beta$ is a DC (difference of two convex functions) function.*

The proof of the proposition above (see Appendix A.11) relies on the fact that under assumptions 6 and 2, $g_{P_n}$ and $\overline{g}_{P_n}^\beta$ are both weakly convex functions, and consequently their difference is a difference of convex functions.

We aim to find $\theta \in \mathbb{R}^m \setminus \{0\}$ that minimizes $\ell_{P_n}^\beta$. A natural approach is to identify a stationary point of $\ell_{P_n}^\beta$ (i.e., a $\theta$ where the subgradient of $\ell_{P_n}^\beta$ contains 0). Despite $\ell_{P_n}^\beta$ having a DC structure, it may still be non-smooth, making the task of finding a stationary point challenging. To address this, we introduce a smoothed version of $\ell_{P_n}^\beta$.

**Definition 2.** *(s-CILO loss)* For all $\lambda \in \mathbb{R}^m$ and $\beta \ge \beta_{\mathcal{H},P_n}$, we denote $g_{P_n}^1 = g_{P_n}$ and $g_{P_n}^2 = \overline{g}_{P_n}^\beta$. For $i \in \{1,2\}$, let

$$M_{P_n}^i(\lambda) := \min_{\theta\in\mathbb{R}^m}(g_{P_n}^i(\theta) + \frac{1}{2}\|\lambda - \theta\|^2),$$

$$\theta_{P_n}^i(\lambda) = \arg\min_{\theta\in\mathbb{R}^m}(g_{P_n}^i(\theta) + \frac{1}{2}\|\theta - \lambda\|^2).$$

*We define a smooth surrogate to the CILO loss, which we call the s-CILO loss $r_{P_n}^\beta(\lambda) := M_{P_n}^1(\lambda) - M_{P_n}^2(\lambda)$. When there is an ambiguity about the value of $\beta$, we denote $\theta_{P_n} := \theta_{P_n}^1$, $\overline{\theta}_{P_n}^\beta := \theta_{P_n}^2$, $M_{P_n} := M_{P_n}^1$, $\overline{M}_{P_n}^\beta := M_{P_n}^2$.*

Following (Sun and Sun, 2021) (proposition 1, page 10), the smooth surrogate above has the following property: for every stationary point $\lambda$ of $r_{P_n}^\beta$, $\overline{\theta}_{P_n}^\beta(\lambda)$ and $\theta_{P_n}(\lambda)$ are equal and are stationary points of $\ell_{P_n}^\beta$. Similar to $\ell_{P_n}^\beta$, s-CILO satisfies the positivity property and the fact that its minimum value is equal to 0 from $\ell_{P_n}^\beta$.

We now have a way to find a stationary point of $\ell_{P_n}^\beta$. A practical way to ensure that the iterates of an optimization algorithm for $\ell_{P_n}^\beta$ do not converge to $\theta = 0$ is to add a constraint $\|\theta\| \ge z$ where $z > 0$. Since the function $\theta \longmapsto z - \|\theta\|$ can be written as a difference of two convex functions, we can find a constrained stationary point to

$r_{P_n}^\beta$ (see (Pang et al., 2017) for tractability results for DC constrained minimization).

The DC structure of our surrogate enables us to find a stationary point of $\ell_{P_n}^\beta$, thus allowing us to approximately minimize the target loss $\ell_P$. Moreover, as we will demonstrate (see Theorem 4 below), for specific choices of the hypothesis set (e.g., a linear hypothesis set), our surrogate has a favorable landscape, meaning that every stationary point of $\ell_{P_n}^\beta$ is a global minimum. Consequently, first-order optimization algorithms can effectively find a global minimum of our surrogate.

From now on, we assume that our hypothesis set $\mathcal{H}$ is linear, meaning there exists a mapping $\Phi : \mathbb{R}^k \longrightarrow \mathbb{R}^{m \times d}$ such that for all $\theta \in \mathbb{R}^m$, $\hat{c}_\theta(x) = \Phi(x)\theta$. In this scenario, Assumption 4 is satisfied if $\Phi(x)$ is full rank almost surely. Additionally, Assumption 6 is inherently satisfied, and Assumption 3 can be replaced with the following:

**Assumption 7.** *For any $x$, the largest singular value of $\Phi(x)$ is bounded above by $B_\Phi$.*

With this special choice of hypothesis set, our surrogate loss and its smoothed version enjoy good landscape properties. In particular, every stationary point of $\ell_{P_n}^\beta$ is a global minimum. We formulate our main landscape result.

**Theorem 4.** $r_{P_n}^\beta$ *is everywhere differentiable. If $\lambda$ is a stationary point of $r_{P_n}^\beta$ such that $\hat{c}_{\overline{\theta}_{P_n}^\beta(\lambda)}(x) \neq 0$ almost surely, then $\overline{\theta}_{P_n}^\beta(\lambda)$ is a global minimum of $\ell_{P_n}^\beta$. In particular, under assumptions 2 and 3, we have for every $\theta \in \mathbb{R}^m$ and $\varepsilon \geq 0$,*

$$\left\| \nabla r_{P_n}^\beta \right\| \leq \varepsilon \implies \ell_{P_n}^\beta(\theta) \leq 8 B_W B_\Phi \varepsilon.$$

A more complete version of this theorem along with its proof is provided in Appendix A.13. Assuming a priori that running gradient descent to optimize our smooth surrogate, we do not fall into the case $\hat{c}_\theta(x) = 0$ almost surely, our practical procedure to optimize $\ell_P$ simply consists of finding a suitable $\beta$ using line search and run gradient descent on $\ell_{P_n}^\beta$.

Now, the only remaining problem to address is the case where the algorithm gives $\theta = 0$.

**Avoiding zero solutions.** Even if we successfully minimize $\ell_{P_n}^\beta$, we could encounter the pathological case where $\theta = 0$, which would fail to ensure that $\hat{c}_\theta(x) \neq 0$ almost surely. In this situation, the conditions for Theorems 1 and 3 would not be satisfied. To prevent this, we leverage the linear structure of the hypothesis set. We begin by stating a proposition that highlights key properties of our surrogate when the hypothesis set is linear.

**Definition 3.** *For a set $V$ in $\mathbb{R}^m$ and element $u \in \mathbb{R}^m$, denote $-V = \{-v, \ v \in V\}$ and $d(u, V) = \min_{v \in V} \|u - v\|$ the $L^2$ distance between $u$ and $V$.*

**Proposition 2.** *Let $\lambda \in \mathbb{R}^m$ and $\beta \geq \beta_{\mathcal{H}, P_n}^\star$, we denote*

$$V_{P_n} := \left\{ \mathbb{E}_{(x,c) \sim P} \left( \Phi^\top(x) w_{P_n}(x) \right), w_{P_n} \in W_{P_n} \right\},$$
$$\overline{V}_{P_n}^\beta := \left\{ \mathbb{E}_{(x,c) \sim P} \left( \Phi^\top(x) w_{P_n}(x) \right), w_{P_n} \in \overline{W}_{P_n}^\beta \right\}.$$

$r_{P_n}^\beta$ *can be rewritten as*

$$r_{P_n}^\beta(\lambda) = \frac{1}{2} \left( d(\lambda, -\overline{V}_{P_n}^\beta)^2 - d(\lambda, -V_{P_n})^2 \right).$$

The proof of this proposition can be found in Appendix A.15.

We observe that $r_{P_n}^\beta$ exhibits an interesting property: recovering a candidate solution $\theta \in \mathbb{R}^m$ that is equal to zero is equivalent to finding $\lambda \in \mathbb{R}^m$ such that $d(\lambda, -\overline{V}_{P_n}^\beta) = d(\lambda, -V_{P_n}) = 0$, and hence, we would like to find a near stationary point for $r_{P_n}^\beta$ such that these distances are non zero. In order to do that, we introduce the following log-barrier surrogate.

**Definition 4.** *(log-CILO loss) Let $\beta \geq \beta_{\mathcal{H}, P}^\star$, we define for every $\theta \in \mathbb{R}^m$,*

$$f_{P_n}^\beta(\lambda) = \log \ d(\lambda, -\overline{V}_{P_n}^\beta) - \log \ d(\lambda, -V_{P_n}). \quad (13)$$

*We call this function the log-CILO loss.*

The function $f_{P_n}^\beta$ inherits the properties we previously observed in $r_{P_n}^\beta$. Specifically, since $d(\lambda, -\overline{V}_{P_n}^\beta) \geq d(\lambda, -V_{P_n})$ for every $\lambda \in \mathbb{R}^m$ (due to the fact that $\overline{V}_{P_n}^\beta \subset V_{P_n}$), and because the logarithm is a non-decreasing function, we have $f_{P_n}^\beta(\lambda) \geq 0$ for every $\lambda \in \mathbb{R}^m$. Furthermore, similar to $r_{P_n}^\beta$, if $\lambda$ is a minimizer of $f_{P_n}^\beta$, then the distances $d(\lambda, -\overline{V}_{P_n}^\beta)$ and $d(\lambda, -V_{P_n})$ are equal. Additionally, it can be shown that any stationary point of $f_{P_n}^\beta$ is also a stationary point of $r_{P_n}^\beta$ (see Theorem 6).

If gradient descent is used to optimize $r_{P_n}^\beta$ and a subsequence of the iterates converges outside the set $-V_{P_n}$ (and hence outside $-\overline{V}_{P_n}^\beta$ as well), we can recover from the limit of this sequence a $\theta \in \mathbb{R}^m$ that is bounded away from zero and serves as a minimizer of $\ell_{P_n}^\beta$. As a result, by Theorems 2 and 3, we successfully obtain a desirable approximate minimizer of $\ell_P$. However, if every limit point of the gradient descent iterates converges within the sets $-V_{P_n}$ and $-V_{P_n}^\beta$, the function $f_{P_n}^\beta$ can be used instead. Gradient descent applied to $f_{P_n}^\beta$ will not yield iterates converging inside the sets $-V_{P_n}$ and $-V_{P_n}^\beta$ due to the presence of log barriers. The following proposition outlines the possible outcomes when running gradient descent on $f_{P_n}^\beta$.

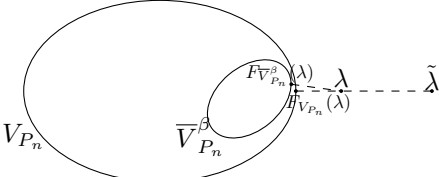

Figure 2. Construction of $\tilde{\lambda}$: when $\lambda$ is close to the sets $\overline{V}_{P_n}^{\beta}$ and $V_{P_n}$, we construct a new approximately stationary point by moving away from $V_{P_n}$ orthogonally to its boundary.

**Proposition 3.** *Consider the backtracking line search gradient descent method applied to the function $f_{P_n}^{\beta}$, ensuring sufficient decrease at each iteration (refer to Algorithm 3.1 in (Nocedal and Wright, 1999)). Suppose the gradient descent sequence has at least one limit point. There are two possible cases:*

1. *The limit point lies strictly outside the interior of $-V_{P_n}$ or $-\overline{V}_{P_n}^{\beta}$;*

2. *The limit point lies on the common boundary of $-V_{P_n}$ and $-\overline{V}_{P_n}^{\beta}$.*

The proof of this proposition is provided in Appendix A.17.

If the gradient sequence $\{\lambda_{i_t}\}$ converges to the common boundary of $-V_{P_n}$ and $-\overline{V}_{P_n}^{\beta}$, the corresponding sequence $\{\overline{\theta}_{P_n}^{\beta}(\lambda_{i_t})\}$ approaches zero, which is not a desirable solution. Fortunately, this issue can be addressed. Since the sequence converges to a limit point on the boundary of $-V_{P_n}$ and $-\overline{V}_{P_n}^{\beta}$ which is also a stationary point of $r_{P_n}^{\beta}$, there must exist some $\lambda_{i_t}$ such that is an $\varepsilon$-stationary point of $r_{P_n}^{\beta}$. We then move $\lambda_{i_t}$ orthogonally away from the boundary of $-V_{P_n}$ (see Figure 5). This yields a new approximate stationary point that is sufficiently distant from both $-V_{P_n}$ and $-V_{P_n}^{\beta}$. As a result, the corresponding $\theta$ is an approximate global minimizer of $\ell_{P_n}^{\beta}$ that is bounded away from zero.

Theorem 6 in Appendix A.16 provides an overview of this procedure, as well as a relationship between the optimality for $f_{P_n}^{\beta}$ and $r_{P_n}^{\beta}$.

# 4. Computational experiments

We make a similar comparison to the one performed by (Hu et al., 2022). In their work, SLO is compared to SPO+, showing that when the hypothesis set is well-specified, SLO achieves a smaller value of $\ell_P$ than SPO+. Conversely, when the model is misspecified, SPO+ performs better. Here, we

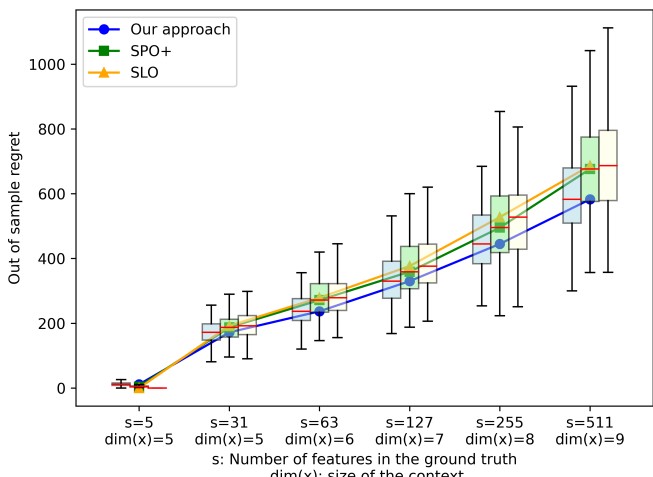

Figure 3. Comparison of SPO+, SLO, and our approach comparison for different levels of misspecification

compare our method with SPO+ and SLO in misspecified setting.

We write the ground truth cost as $c(x) = \theta^{\star}\phi(x)$ for some matrix $\theta^{\star} \in \mathbb{R}^{20 \times s}$ and for all $x \in \mathbb{R}^k$, where $\phi(x) \in \mathbb{R}^s$. Furthermore, the predictors in the hypothesis set take the form $\hat{c}_{\theta}(x) = \theta\tilde{\phi}(x)$ for any $\theta \in \mathbb{R}^{20 \times 5}$ and for all $x \in \mathbb{R}^k$, where $\tilde{\phi}(x) \in \mathbb{R}^5$ is equal to $\phi(x)$ truncated to its first 5 coordinates. When $s = 5$, the hypothesis set is well-specified, and when $s$ grows, the misspecification level of the hypothesis set grows as well. In this setting, the hypothesis set is linear (see Appendix A.18 for more details). To optimize $\ell_{P_n}^{\beta}$, we ran gradient descent on its surrogate loss $r_{P_n}^{\beta}$ (we did not need to optimize the surrogate $f_{P_n}^{\beta}$ because the iterates did not converge to 0). We chose $\beta$ by line search. We used $\beta_{\min,P} = \mathbb{E}_{(x,c) \sim P_n} \left( c^{\top} w(c) \right)$ as a lower bound to $\beta$, and $\beta_{\text{SPO+}} = \mathbb{E}_{(x,c) \sim P_n} \left( c^{\top} w \left( \hat{c}_{\theta_{\text{SPO+}}^{\star}}(x) \right) \right)$ where $\theta_{\text{SPO+}}^{\star}$ is the solution obtained by optimizing the SPO+ loss. For every value of $s$, we tested 96 evenly spaced values of $\beta$ in the interval $[\beta_{\min,P}, \beta_{\text{SPO+}}]$, and picked $\beta$ yielding the solution with the best decision performance. The optimal $\beta$ is likely to fall into (or near) the interval $[\beta_{\min,P}, \beta_{\text{SPO+}}]$, and hence for at least one of the values of $\beta$ that we test, $\ell_P^{\beta}$ enjoys good optimality guarantees, which ensures that the solution we obtain yields a small value for $\ell_P$. For every value of $s$, we run 96 experiments to get the distribution of the testing loss $\ell_P$ for every method, and get results in the box plots in Figure 3. We also include the same plot with relative regret instead of absolute regret in Appendix D.

In Figure 3, we plot the regret yielded by SLO, SPO+ and our approach, i.e. the value of $\ell'_P(\theta) = \ell_P(\theta) - \beta_{\min,P} \geq 0$. We observe that in the well-specified setting ($s = 5$), SLO

performs best among the other two methods, whereas in the misspecified setting ($s > 5$), the more the misspecification level grows, the more our method performs better than SLO and SPO+, and SLO performs the worst. This is coherent with results in (Hu et al., 2022) and (Elmachtoub et al., 2023) in one hand, but also with our consistency and generalization results in the other hand (Theorems 1 and 2).

## 5. Conclusion

This paper presents a novel approach to addressing model misspecification in contextual optimization. State-of-the-art methods optimize the decision cost effectively only when the hypothesis set is well-specified, leaving the misspecified case largely unresolved. Our surrogate loss function successfully optimizes the decision cost and retrieves a good cost predictor in terms of decision performance from the hypothesis set in reasonable time. Despite its non-convexity and non-smoothness, we exploit its structure as a difference of convex functions, enabling optimization through smoothing. We theoretically and experimentally demonstrate that our approach outperforms state-of-the-art methods in misspecified settings. To our knowledge, this is the first approach to provably optimize the decision cost $\ell_P$ under misspecification. While almost all of our results only require generic boundedness and smoothness assumptions, global optimality for our surrogate loss holds for linear hypothesis sets, although we believe it extends to a broader class of predictors and contextual optimization problems beyond the case where the objective to optimize is linear. We have validated our method on synthetic data and plan further experiments on real-world datasets for comparison with existing methods.

## Impact Statement

This paper introduces a tractable method for contextual optimization under model misspecification, with theoretical guarantees and improved decision performance. The contribution is purely methodological; we do not anticipate any broader societal or ethical impact beyond this technical advance.

## Acknowledgements

This work has been partly funded by AFOSR grant FA9550-19-1-0263. The author Jiawei Zhang is supported by MIT postdoc fellowship for engineering excellences. All computational experiments were run on the MIT SuperCloud (Reuther et al., 2018).

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

# A. Appendix

### A.1. More details about Example 1

For any predictor $\hat{c}$, we use the following abuse of notation

$$\ell_P(\hat{c}) = \mathbb{E}_{(x,c)\sim P}\left(\max_{w\in W^\star(\hat{c}(x))} c^\top w\right),$$

$$\ell_{\text{SPO+}}(\hat{c}) := \mathbb{E}_{(x,c)\sim P}\left(\max_{w\in W}(c^\top w - 2(\hat{c}^\top w - \hat{c}^\top w(c)))\right)$$

$$\ell_{\text{SLO}}(\hat{c}) = \sqrt{\mathbb{E}_{(x,c)\sim P}\left(\|c - \hat{c}(x)\|^2\right)}.$$

Notice that whereas $\ell_P(\hat{c}_1) = -\frac{1}{2}$ and $\ell_P(\hat{c}_2) = 0 > \ell_P(\hat{c}_1)$, we have $\ell_{\text{SPO+}}(\hat{c}_1) = \frac{3}{4}$, $\ell_{\text{SPO+}}(\hat{c}_2) = \frac{2}{3} < \ell_{\text{SPO+}}(\hat{c}_1)$, $\ell_{\text{SLO}}(\hat{c}_1) = \frac{7}{8}$, $\ell_{\text{SLO}}(\hat{c}_2) = \frac{7}{12} < \ell_{\text{SLO}}(\hat{c}_1)$. Although $\hat{c}_2$ leads to suboptimal decisions while $\hat{c}_1$ results in optimal decisions, both SLO and SPO+ appear to favor $\hat{c}_2$.

### A.2. Equivalence of the target loss in binary classification

***Proof.*** For every $\hat{c} : \mathbb{R}^k \longrightarrow \mathbb{R}^d$, we denote

$$\ell_P(\hat{c}) = \mathbb{E}_{(x,c)\sim P}\left(\min_{w\in W^\star(\hat{c}(x))} c^\top w\right).$$

We have for every $\hat{c} : \mathbb{R}^k \longrightarrow \mathbb{R}^d$, for every $x \in \mathbb{R}^k$, if $\text{sign}(\hat{c}(x)) = c$, then $W^\star(\hat{c}(x)) = W^\star(c) = \{c/2\}$. Consequently, we have $\min_{w\in W^\star(\hat{c}(x))} c \cdot w = \frac{1}{2}$. If $\text{sign}(\hat{c}(x)) \neq c$, then $\min_{w\in W^\star(\hat{c}(x))} c \cdot w = -\frac{1}{2}$. Consequently, we have for any $x \in \mathbb{R}^k$,

$$\frac{1}{2} + \min_{w\in W^\star(\hat{c}(x))} c \cdot w = \begin{cases} 1 & \text{if } \text{sign}(\hat{c}(x)) = c \\ 0 & \text{else.} \end{cases}$$

This directly gives

$$\ell_P(\hat{c}) = \mathbb{E}_{(x,c)\sim P}\left(\mathbb{1}_{\text{sign}(\hat{c}(x))=c}\right) + \frac{1}{2}.$$

$\square$

### A.3. Previous approaches to Integrated Learning and Optimization

A more comprehensive review of contextual optimization approaches can be found in the survey by (Sadana et al., 2024).

**Directly optimizing the target loss.** One of the first instances of the ILO method can be traced back to (Donti et al., 2017) who provide a practical way to differentiate the target loss $\ell_P$ under some regularity conditions. Similarly, others have attempted to directly minimize $\ell_P$ using some estimation of its gradient, such as unrolling (Domke, 2012; Monga et al., 2021), which consists of keeping track of operations while running gradient descent in order to differentiate the final gradient descent iterate as a function of the model parameters, and implicit differentiation (Amos and Kolter, 2017; Agrawal et al., 2019; Sun et al., 2022; McKenzie et al., 2023). However, $\ell_P$ is non-differentiable in general, and even if it were, there is no guarantee that it would be convex. This implies that gradient-based algorithms mentioned above do not guarantee convergence to optimum of the target loss. In contrast, our approach consists of minimizing a tractable smooth surrogate that has the same set of minimizers as $\ell_P$. This allows us to avoid the challenge coming from the lack of regularity properties for $\ell_P$.

**Optimizing a surrogate loss.** Another increasingly popular approach, which is most relevant to ours, is optimizing the target loss using a smooth convex surrogate loss. In (Elmachtoub and Grigas, 2022), a new convex surrogate function, named SPO+. Minimizing SPO+ is proven to also minimize the target loss when the hypothesis set is well-specified, but no

consistency results are provided when the chosen predictor class is misspecified. Furthermore, SPO+ seems to outperform SLO (i.e. yield better decision performance) when $\mathcal{H}$ is misspecified, since SLO only focuses on the accuracy of the prediction step, but completely disregards the performance in the optimization step, whereas ILO specifically focuses on the decision performance, and when the hypothesis set is misspecified, maximizing the prediction accuracy of the underlying cost will not necessarily result in good decision performance. Moreover, when $\mathcal{H}$ is well-specified, SLO methods outperform ILO (Hu et al., 2022; Elmachtoub and Grigas, 2022). (Hu et al., 2022) give theoretical and experimental evidence showing that classical SLO methods generalize better than SPO+ when the hypothesis set is well-specified. In fact, evidence in (Elmachtoub et al., 2023) suggests that in the well-specified case, SLO might likely have better performance than ILO approaches, and such a behavior is inverted in the misspecified case. Most previous works have not theoretically considered the misspecified case. In (Huang and Gupta, 2024), a new surrogate is introduced based on an approximation of the directional derivative of $\ell_P$, and is shown to be theoretically consistent with $\ell_P$. However, only local optimality results are provided for this surrogate, whereas in our work, we provide global optimality guarantees.

Another alternative is the surrogate introduced by (Sun et al., 2023), which attempts to maximize the nonbasic reduced costs (when the cost is taken equal to the predicted cost) of past realizations of ground truth optimal decisions. Such a method does not require the knowledge of historical costs, but only solutions of previously seen linear programs. In order for this surrogate to be consistent, authors assume that the chosen hypothesis set is well-specified in the sense of our definition, i.e. $\gamma_{\text{dec}}(\mathcal{H}) = 0$. On one hand, this assumption is weaker than the usual definition of well-specification, which requires that the hypothesis set contains the ground truth predictor. On the other hand, this surrogate is only designed for linear objectives with linear constraints. Our surrogate only requires the set of feasible decisions $W$ to be convex and bounded. In a related paper, (Liu et al., 2021) use a neural network structure in an inventory management problem to learn a mapping which provides the optimal merchandise order quantity and order time, and give theoretical guarantees in the well-specified setting. Other surrogates have been considered in the literature (Loke et al., 2022; Jeong et al., 2022; Kallus and Mao, 2023), which despite having good practical performance benefits, do not seem to theoretically tackle the misspecified case as opposed to our work, though practical toolkits such as PyEPO (Tang and Khalil, 2024) provide implementations of some of these approaches.

### A.4. Prediction misspecification and decision misspecification

In the litterature, a classical metric to quantify the level of prediction misspecification of a given hypothesis set $\mathcal{H}$ is the $L^1$ or $L^2$ norm of the distance between the parameter (or the function) one seeks to estimate and the chosen hypothesis set. In our setting, it can be written as

$$\gamma_{\text{pred}}(\mathcal{H}) = \sqrt{\min_{\hat{c} \in \mathcal{H}} \mathbb{E}_{(x,c) \sim P}\left((\hat{c}(x) - c(x))^2\right)}.$$

This metric has been previously adopted in statistics and contextual bandits (Foster et al., 2020), (Krishnamurthy et al., 2021). We argue that $\gamma_{\text{pred}}(\mathcal{H})$ can be a good metric for evaluating the effect of misspecification for prediction problems, but not for contextual optimization problems in which parameter estimates enter in downstream optimization problem. This can be seen by noting that in our setting, the best achievable cost should not change when multiplying the elements of $\mathcal{H}$ by a positive constant, whereas $\gamma_{\text{pred}}(\mathcal{H})$ is not invariant to such a transformation. To address this, we introduce a new metric to quantify misspecification in contextual optimization.

**Definition 5.** *We define the decision misspecification gap as*

$$\gamma_{dec}(\mathcal{H}) = \min_{\hat{c} \in \mathcal{H}} \mathbb{E}_{(x,c) \sim P}\left(\max_{w \in W^\star(\hat{c}(x))} c(x)^\top w\right) - \mathbb{E}\left(\min_{w \in W} c(x)^\top w\right).$$

*The function class $\mathcal{H}$ is decision-well-specified if $\gamma_{dec}(\mathcal{H}) = 0$, and decision-misspecified otherwise.*

First, note that the first term is the smallest possible value of the target loss when choosing a cost predictor from $\mathcal{H}$, and the second term is the smallest possible value of the target loss in the case where $\mathcal{H}$ contains the ground truth cost predictor. Hence, it follows to conclude that $\gamma_{\text{dec}}(\mathcal{H})$ captures the optimality gap (in terms of decision performance) caused by model misspecification. Second, notice that if $\mathcal{H}$ is decision-misspecified, i.e. $\gamma_{\text{dec}}(\mathcal{H}) > 0$, then necessarily it is prediction-misspecified, i.e. $\gamma_{\text{pred}}(\mathcal{H}) > 0$. Third, $\gamma_{\text{dec}}(\mathcal{H})$ is clearly invariant when multiplying the cost predictors by a constant. In Example 1, that we have $\gamma_{\text{pred}}(\mathcal{H}) > 0$ but $\gamma_{\text{dec}}(\mathcal{H}) = 0$. This means that while SPO+ and SLO are consistent when the hypothesis set is prediction-well-specified, their consistency property no longer holds even when the hypothesis set is decision-well-specified in general.

For a given cost predictor $\hat{c}$, denoting $\gamma_{\text{dec}}(\hat{c}) = \mathbb{E}_{(x,c)\sim P} \left( \max_{w\in W^\star(\hat{c}(x))} c^\top w \right) - \mathbb{E} \left( \min_{w\in W} c^\top w \right)$ and $\gamma_{\text{pred}}(\hat{c}) = \sqrt{\mathbb{E}_{(x,c)\sim P}\left( (\hat{c}(x) - c(x))^2 \right)}$, we can see that SLO focuses on bringing $\gamma_{\text{pred}}(\hat{c})$ as close as possible to $\gamma_{\text{pred}}(\mathcal{H})$, whereas it is more natural to focus on bringing $\gamma_{\text{dec}}(\hat{c})$ as close to $\gamma_{\text{dec}}(\mathcal{H})$ as possible, since small prediction error does not necessarily mean good decision performance in the misspecified case. In the other hand, even if current ILO methods such as SPO+ aim to bring $\gamma_{\text{dec}}(\hat{c})$ as close to $\gamma_{\text{dec}}(\mathcal{H})$, it is unclear whether they can do so in the misspecified case, whereas our surrogate loss's optimality gap dominates $\gamma_{\text{dec}}(\hat{c})$ (see Theorem 3).

### A.5. Sufficient condition for uniqueness assumption to hold when $W$ is a polyhedron

**Proposition 4.** *If the context variable $x$ has a continuous probability distribution and the mapping $x \longmapsto \hat{c}_\theta(x)$ is a nonzero analytic function for any $\theta \neq 0$, then Assumption 1 holds for any polyhedron $W$ that satisfies $\hat{c}_\theta(x)^\top v \neq 0$ for any $\theta \in \mathbb{R}^m \setminus \{0\}$ and any nonzero $v \in W$.*

**Proof.** Let $\{S_i\}_{i\in I}$ be all the faces of $W$, where $I$ is a finite index set. Let $v_i$ be an arbitrary tangent direction of $S_i$ and let $\mathcal{V} = \{v_i\}_{i\in I}$ be a finite set. Let $U$ be the set containing all $u$ such that the solution to the linear program $\min_{w\in W} u^\top w, u \in U$ is not unique. Then any $u \in U$ satisfies $u^\top v_i = 0$ for some $i$. Therefore, $\hat{c}_\theta(x) \in U$ if $\hat{c}_\theta(x)^\top v_i = 0$ for some $i$. Let $X_0 := \{x \in \mathbb{R}^k \mid \hat{c}_\theta(x) \in U\}$ and $X_i := \{x \in \mathbb{R}^k \mid \hat{c}_\theta(x)^\top v_i = 0\}$. We have $X_0 \subseteq \bigcup_{i\in I} X_i$. By the assumption in the Proposition, $\hat{c}_\theta(x)^\top v_i$ is a nonzero analytic function and then by (Mityagin, 2015), $X_i$ is a zero-measure set. Therefore, $X_0 = \bigcup_{i\in I} X_i$ is a zero-measure set since $I$ is finite. Finally, since $X_0$ is of measure zero and $x$ is a continuous random variable, $x$ lies outside of $X_0$ almost surely, i.e. the solution to $\min_{w\in W} \hat{c}_\theta(x)^\top w$ is unique almost surely. $\square$

### A.6. Proof of Proposition 2.1

In the binary classification setting, when $\mathcal{H}$ is decision-well-specified, we have $\beta^\star_{\mathcal{H},P} = -\frac{1}{2}$. For any $\hat{c} : \mathbb{R}^k \longrightarrow \mathbb{R}^d$, when $\beta = \beta^\star_{\mathcal{H},P}$,

$$
\begin{aligned}
\ell^\beta_P(\hat{c}) &= - \max_{\overline{w}^\beta_P \in \overline{W}^\beta_P} \mathbb{E}_{(x,c)\sim P} \left( \hat{c}(x) \cdot \overline{w}^\beta_P(x) \right) + \max_{w_P \in W_P} \mathbb{E}_{(x,c)\sim P} \left( \hat{c}(x) \cdot w_P(x) \right) \\
&= - \max_{\substack{\mathbb{E}_{(x,c)\sim P}\left(c\cdot\overline{w}^\beta_P(x)\right)\geq\frac{1}{2} \\ \overline{w}^\beta_P \in W_P}} \mathbb{E}_{(x,c)\sim P} \left( \hat{c}(x) \cdot \overline{w}^\beta_P(x) \right) + \frac{1}{2}\mathbb{E}_{(x,c)\sim P} \left( |\hat{c}(x)| \right) \\
&= -\frac{1}{2}\mathbb{E}_{(x,c)\sim P} \left( \hat{c}(x) \cdot c \right) + \frac{1}{2}\mathbb{E}_{(x,c)\sim P} \left( |\hat{c}(x)| \right) \\
&= \frac{1}{2}\mathbb{E}_{(x,c)\sim P} \left( |\hat{c}(x)| \left( 1 - \text{sign}(\hat{c}(x)) \cdot c \right) \right) \\
&= \mathbb{E}_{(x,c)\sim P} \left( |\hat{c}(x)| \, \mathbb{1}_{\text{sign}(\hat{c}(x))\neq c} \right).
\end{aligned}
$$

### A.7. Proof of Theorem 1

**Proof.** We denote $\beta^\star_{\mathcal{H},P} = \min_{\theta\in\mathbb{R}^m} \ell_P(\theta)$, $\beta_{\max,P} = \mathbb{E}_{(x,c)\sim P}(\max_{w\in W} c^\top w)$, and $\beta_{\min,P} = \mathbb{E}_{(x,c)\sim P}\left(\min_{w\in W} c^\top w\right)$. Let $\beta \in \mathbb{R}$. We first study the property $\ell_P(\theta) \leq \beta$ when $\beta$ satisfies $\beta^\star_{\mathcal{H},P} \leq \beta < \beta_{\max,P}$ and $\hat{c}_\theta(x) \neq 0$ almost surely. The inequality $\beta^\star_{\mathcal{H},P} \leq \beta$ ensures that the condition $\ell_P(\theta) \leq \beta$ is feasible, and the inequality $\beta < \beta_{\max,P}$ ensures that the condition $\ell_P(\theta) \leq \beta$ is not trivial. Notice that if we take $\beta = \beta^\star_{\mathcal{H},P}$, any $\theta \in \mathbb{R}^m$ satisfying $\ell_P(\theta) \leq \beta$ is a minimizer of $\ell_P(\theta)$. When Assumption 1 holds, and when $\hat{c}_\theta(x) \neq 0$ almost surely, $W^\star(\hat{c}_\theta(x))$ contains only one element for almost every $x$. Hence, we denote $w(\hat{c}_\theta(x))$ to be the unique solution in $W^\star(\hat{c}_\theta(x))$ when $W^\star(\hat{c}_\theta(x))$ contains only one element. The condition $\ell_P(\theta) \leq \beta$ can be rewritten as $\mathbb{E}_{(x,c)\sim P}(c^\top w(\hat{c}_\theta(x))) \leq \beta$ when $\hat{c}_\theta(x) \neq 0$ almost surely. For a given $\theta \in \mathbb{R}^m$ and $w_P \in W_P$, if $w_P$ satisfies

$$
w_P \in \arg \min_{w_P \in W_P} \mathbb{E}_{(x,c)\sim P}(\hat{c}_\theta(x)^\top w_P(x)), \tag{14}
$$

then we have, $w_P(x) = w(\hat{c}_\theta(x))$ almost surely, as for each $x$, the unique minimizer of $\hat{c}_\theta(x)^\top w$ over $w \in W$ is given by $w(\hat{c}_\theta(x))$, according to Assumption 1.

If $\theta \in \mathbb{R}^m$ satisfies $\ell_P(\theta) \leq \beta$, and $w_P$ is the measurable mapping satisfying condition (14) (and hence $w_P(x) = w(\hat{c}_\theta(x))$ almost surely), then we have $\mathbb{E}_{(x,c)\sim P} \left( c^\top w_P(x) \right) \leq \beta$. This suggests that adding the linear constraint

$\mathbb{E}_{(x,c)\sim P}\left(c^\top w_P(x)\right) \leq \beta$ to (14) does not change the set of minimizers of $w_P \longmapsto \mathbb{E}_{(x,c)\sim P}(\hat{c}_\theta(x)^\top w_P(x))$ when $\ell_P(\theta) \leq \beta$. Defining $\overline{W}_P^\beta = \left\{\overline{w}_P \in W_P,\ \mathbb{E}_{(x,c)\sim P}\left(c^\top \overline{w}_P(x)\right) \leq \beta\right\}$, our latter statement means that if $\ell_P(\theta) \leq \beta$, the two optimization problems

$$\min_{w_P \in W_P} \mathbb{E}_{(x,c)\sim P}\left(\hat{c}_\theta(x)^\top w_P(x)\right) \tag{15}$$

and

$$\min_{\overline{w}_P \in \overline{W}_P^\beta} \mathbb{E}_{(x,c)\sim P}\left(\hat{c}_\theta(x)^\top \overline{w}_P(x)\right) \tag{16}$$

would result in identical values of the objective function.

Let $\beta \in \mathbb{R}$ such that $\beta_{\mathcal{H},P}^\star \leq \beta < \beta_{\max,P}$ and $\theta \in \mathbb{R}^m$ satisfying $\hat{c}_\theta(x) \neq 0$ almost surely. Assume that $\theta$ is optimal for $\ell_P^\beta$, i.e. $\ell_P^\beta(\theta) = 0$. Let $\overline{w}_P(x)$ be the solution of (16). Then $\ell_P^\beta(\theta) = 0$ implies

$$\mathbb{E}_{(x,c)\sim P}\left(\hat{c}_\theta(x)^\top \overline{w}_P(x) - \hat{c}_\theta(x)^\top w\left(\hat{c}_\theta(x)\right)\right) = 0.$$

Since the random variable $\hat{c}_\theta(x)^\top \overline{w}_P(x) - \hat{c}_\theta(x)^\top w\left(\hat{c}_\theta(x)\right)$ is positive almost surely, the equality above implies that $\hat{c}_\theta(x)^\top \overline{w}_P(x) - \hat{c}_\theta(x)^\top w\left(\hat{c}_\theta(x)\right) = 0$ almost surely, i.e. $\overline{w}_P$ is also a minimizer of $w_P \longmapsto E_{(x,c)\sim P}\left(\hat{c}_\theta(x)^\top w_P(x)\right)$, and hence under Assumption 1, we have $\overline{w}_P(x) = w(\hat{c}_\theta(x))$ almost surely. This finally gives

$$\ell_P(\theta) = \mathbb{E}_{(x,c)\sim P}(c^\top w\left(\hat{c}_\theta(x)\right)) = \mathbb{E}_{(x,c)\sim P}(c^\top \overline{w}_P(x)) \leq \beta.$$

$\square$

### A.8. More details about Assumption 5

We first mention the assumption in (Hu et al., 2022) which enables the authors to get good generalization guarantees for SLO in the well-specified case. Note that we will not be making this assumption, but we mention it just for reference.

**Assumption 8.** *Assume that $W$ is a polyhedron. We denote $W^\angle$ the set of extreme points of $W$. For a given context $x \in \mathbb{R}^k$, we denote*

$$\Delta(x) = \begin{cases} \min_{w \in W^\angle \setminus W^\star(c(x))} c(x)^\top w - \min_{w \in W^\angle} c(x)^\top w & \text{if } W^\star(c(x)) \neq W \\ 0 & \text{else.} \end{cases}$$

*Assume that for some $\alpha, \gamma \geq 0$,*

$$\forall t > 0,\ \mathbb{P}(0 < \Delta(x) \leq t) \leq \left(\frac{\gamma t}{B_W}\right)^\alpha.$$

We provide further justification of the remarks following Assumption 5. When $W$ is a polyhedron, denoting for $\theta \neq 0$, $\underline{w}(\theta, x) = \arg\min_{w \in W^\angle \setminus \{w(\hat{c}_\theta(x))\}} \hat{c}_\theta(x)^\top w$, and

$$\cos\left(\omega(\theta, x)\right) = \frac{\hat{c}_\theta(x)^\top (\underline{w}(\theta, x) - w(\hat{c}_\theta(x)))}{\|\hat{c}_\theta(x)\|\,\|\underline{w}(\theta, x) - w(\hat{c}_\theta(x))\|},$$

Assumption 5 can be rewritten as

$$\mathbb{P}\left(\|\hat{c}_\theta(x)\|\,\|\underline{w}(\theta, x) - w(\hat{c}_\theta(x))\|\cos(\omega(\theta, x)) \leq \|\theta\|\,t\right) \leq \left(\frac{\gamma t}{B_W}\right)^\alpha. \tag{17}$$

We do not need to focus on the term $\|\underline{w}(\theta, x) - w(\hat{c}_\theta(x))\|$, as it is both upper bounded and bounded away from zero. Therefore, inequality (17) is equivalent to $\hat{c}_\theta(x)$ having a norm that is bounded away from zero with high probability, and an angle $\omega(\theta, x)$ that is bounded away from $\frac{\pi}{2}$ when $\theta$ is bounded away from 0. In other words, $\hat{c}_\theta(x)$ is likely to have a direction that is not too close to being perpendicular to one of the faces of the polyhedron $W$. In other words, the probability of this direction to be falling within one of the red cones shown in Figure 1 and 4 decays to 0 as the cones get more narrow. When $\hat{c}_\theta(x)$ follows a continuous distribution, such a property on the direction of $\hat{c}_\theta(x)$ is reasonable.

The key idea of the proof of Theorem 3 relies on the following sensitivity inequality resulting from Assumption 5 which holds for every $t \geq 0$ and mapping $w \in W_P$:

$$\mathbb{E}_{(x,c)\sim P}\left(\left|\hat{c}_\theta(x)^\top w(\hat{c}_\theta(x)) - \hat{c}_\theta(x)^\top w(x)\right|\right) \geq tA_1\|\theta\|\,\mathbb{E}_{(x,c)\sim P}\left(\|w(\hat{c}_\theta(x)) - w(x)\|\right) - A_2\|\theta\|\,t^\alpha,$$

where $A_1, A_2$ are positive constants. The left-hand side equals $\ell_P^\beta(\theta)$ for a well-chosen $w \in W_P$, while the right-hand side is directly related to $\ell_P(\theta) - \beta$. Optimizing over $t$ in the right-hand side yields the desired inequality. By omitting the term $A_2|\theta|t^\alpha$ and the dependence of the right-hand side on $t$, it is possible to obtain a stronger bound that does not require $W$ to be a polyhderon. Specifically, if we assume the existence of $B_s > 0$ such that for every $\theta \in \mathbb{R}^m$ and every mapping $w \in W_P$,

$$\mathbb{E}_{(x,c)\sim P}\left(\left|\hat{c}_\theta(x)^\top w(\hat{c}_\theta(x)) - \hat{c}_\theta(x)^\top w(x)\right|\right) \geq B_s\|\theta\|\,\mathbb{E}_{(x,c)\sim P}\left(\|w(\hat{c}_\theta(x)) - w(x)\|\right), \tag{18}$$

the bound $\ell_P(\theta) \leq \beta + O\left(\ell_P^\beta(\theta)\right)$ holds when $\|\theta\|$ is bounded away from $0$. This bound is stronger since $1 - \frac{1}{1+\alpha} < 1$ for every $\alpha > 0$ and does not require $W$ to be a polyhedron. This assumption is also reasonable when $P$ is continuous.

### A.9. Proof of Theorem 2

*Proof.* We proceed in two steps.

**Step 1.** We first prove the Lipschitz continuity of the generalization, i.e. that there exists a constant $A \geq 0$ such that for every $\theta, \theta' \in \mathbb{R}^m$,

$$\left|\ell_{P_n}^\beta(\theta) - \ell_P^\beta(\theta) - \left(\ell_{P_n}^\beta(\theta') - \ell_P^\beta(\theta')\right)\right| \leq A\|\theta - \theta'\|. \tag{19}$$

For every $\theta \in \mathbb{R}^m$, using Lagrangian duality,

$$\ell_{P_n}^\beta(\theta) = \min_{\overline{w}_{P_n}^\beta \in \overline{W}_{P_n}^\beta} \mathbb{E}_{(x,c)\sim P_n}\left(\hat{c}_\theta(x)^\top \overline{w}_{P_n}^\beta(x)\right) - \min_{w_{P_n} \in W_{P_n}} \mathbb{E}_{(x,c)\sim P_n}\left(\hat{c}_\theta(x)^\top w_{P_n}(x)\right) \tag{20}$$

$$= \min_{\overline{w}_{P_n}^\beta \in W_{P_n}} \max_{y\geq 0} \mathbb{E}_{(x,c)\sim P_n}\left(\hat{c}_\theta(x)^\top \overline{w}_{P_n}^\beta(x)\right) + y\left(\mathbb{E}_{(x,c)\sim P_n}\left(c(x)^\top w_{P_n}^\beta(x)\right) - \beta\right) \tag{21}$$

$$- \min_{w_{P_n} \in W_{P_n}} \mathbb{E}_{(x,c)\sim P_n}\left(\hat{c}_\theta(x)^\top w_{P_n}(x)\right) \tag{22}$$

$$= \max_{y\geq 0} \min_{\overline{w}_{P_n}^\beta \in W_{P_n}} \mathbb{E}_{(x,c)\sim P_n}\left((\hat{c}_\theta(x) + yc(x))^\top \overline{w}_{P_n}^\beta(x)\right) - y\beta - \min_{w_{P_n} \in W_{P_n}} \mathbb{E}_{(x,c)\sim P_n}\left(\hat{c}_\theta(x)^\top w_{P_n}(x)\right). \tag{23}$$

In (23), we have switched the min and the max because of strong duality, since the objective function we are optimizing is linear. For a given $\theta \in \mathbb{R}^m$, we denote $y^\star(\theta)$ to be the optimal dual variable corresponding to $\theta$ in the minimization problem above. Let $\theta, \theta' \in \mathbb{R}^m$ and $D \geq \max(\|\theta\|, \|\theta'\|)$. We have

$$\left|\ell_{P_n}^\beta(\theta) - \ell_P^\beta(\theta) - \left(\ell_{P_n}^\beta(\theta') - \ell_P^\beta(\theta')\right)\right| \leq \left|\ell_{P_n}^\beta(\theta) - \ell_{P_n}^\beta(\theta')\right| + \left|\ell_P^\beta(\theta) - \ell_P^\beta(\theta')\right|.$$

We now bound the two terms above on the right side of the inequality. We have

$$\left|\ell_{P_n}^\beta(\theta) - \ell_{P_n}^\beta(\theta')\right| = \left|\max_{y\geq 0} \min_{\overline{w}_{P_n}^\beta \in W_{P_n}} \mathbb{E}_{(x,c)\sim P_n}\left((\hat{c}_\theta(x) + yc(x))^\top \overline{w}_{P_n}^\beta(x)\right) - y\beta - \min_{w_{P_n} \in W_{P_n}} \mathbb{E}_{(x,c)\sim P_n}\left(\hat{c}_\theta(x)^\top w_{P_n}(x)\right)\right.$$

$$\left. - \max_{y\geq 0} \min_{\overline{w}_{P_n}^\beta \in W_{P_n}} \mathbb{E}_{(x,c)\sim P_n}\left((\hat{c}_{\theta'}(x) + yc(x))^\top \overline{w}_{P_n}^\beta(x)\right) - y\beta - \min_{w_{P_n} \in W_{P_n}} \mathbb{E}_{(x,c)\sim P_n}\left(\hat{c}_{\theta'}(x)^\top w_{P_n}(x)\right)\right|$$

$$= \left|\min_{\overline{w}_{P_n}^\beta \in W_{P_n}} \mathbb{E}_{(x,c)\sim P_n}\left((\hat{c}_\theta(x) + y^\star(\theta)c(x))^\top \overline{w}_{P_n}^\beta(x)\right) - y^\star(\theta)\beta\right.$$

$$- \min_{w_{P_n} \in W_{P_n}} \mathbb{E}_{(x,c)\sim P_n}\left(\hat{c}_\theta(x)^\top w_{P_n}(x)\right)$$

$$- \min_{\overline{w}_{P_n}^\beta \in W_{P_n}} \mathbb{E}_{(x,c)\sim P_n}\left((\hat{c}_{\theta'}(x) + y^\star(\theta')c(x))^\top \overline{w}_{P_n}^\beta(x)\right)$$

$$\left.-y^\star(\theta')\beta - \min_{w_{P_n} \in W_{P_n}} \mathbb{E}_{(x,c)\sim P_n}\left(\hat{c}_{\theta'}(x)^\top w_{P_n}(x)\right)\right|.$$

In the left hand side of the inequality above, we can see that if $\ell_{P_n}^\beta(\theta) \geq \ell_{P_n}^\beta(\theta')$, then replacing $y^\star(\theta')$ by $y^\star(\theta)$ gives a larger term than the one above, and when $\ell_{P_n}^\beta(\theta) \leq \ell_{P_n}^\beta(\theta')$, replacing $y^\star(\theta)$ by $y^\star(\theta')$ gives a larger term than the one above as well. Let $\tilde{y} \in \{y^\star(\theta), y^\star(\theta')\}$ such that replacing $y^\star(\theta)$ and $y^\star(\theta')$ by $\tilde{y}$ makes the expression above larger. We have, by replacing $y^\star(\theta)$ and $y^\star(\theta')$ by $\tilde{y}$ in the inequality above, we get

$$
\begin{aligned}
\left| \ell_{P_n}^\beta(\theta) - \ell_{P_n}^\beta(\theta') \right| \leq \Bigg| &\min_{\overline{w}_{P_n}^\beta \in W_{P_n}} \mathbb{E}_{(x,c)\sim P_n} \left( (\hat{c}_\theta(x) + \tilde{y}c(x))^\top \overline{w}_{P_n}^\beta(x) \right) - \min_{w_{P_n} \in W_{P_n}} \mathbb{E}_{(x,c)\sim P_n} \left( \hat{c}_\theta(x)^\top w_{P_n}(x) \right) \\
&- \min_{\overline{w}_{P_n}^\beta \in W_{P_n}} \mathbb{E}_{(x,c)\sim P_n} \left( (\hat{c}_{\theta'}(x) + \tilde{y}c(x))^\top \overline{w}_{P_n}^\beta(x) \right) + \min_{w_{P_n} \in W_{P_n}} \mathbb{E}_{(x,c)\sim P_n} \left( \hat{c}_{\theta'}(x)^\top w_{P_n}(x) \right) \Bigg| \\
= \Bigg| &\mathbb{E}_{(x,c)\sim P_n} \left( \min_{w\in W} (\hat{c}_\theta(x) + \tilde{y}c(x))^\top w \right) - \mathbb{E}_{(x,c)\sim P_n} \left( \min_{w\in W} \hat{c}_\theta(x)^\top w \right) \\
&- \mathbb{E}_{(x,c)\sim P_n} \left( \min_{w\in W} (\hat{c}_{\theta'}(x) + \tilde{y}c(x))^\top w \right) + \mathbb{E}_{(x,c)\sim P_n} \left( \min_{w\in W} \hat{c}_{\theta'}(x)^\top w \right) \Bigg| \\
= &\left| f_{\tilde{y},P_n}(\theta) - f_{\tilde{y},P_n}(\theta') \right|,
\end{aligned}
$$

where for every $\theta \in \mathbb{R}^m$,

$$
f_{\tilde{y},P_n}(\theta) = \mathbb{E}_{(x,c)\sim P_n} \left( \min_{w\in W} \overbrace{(\hat{c}_\theta(x) + \tilde{y}c(x))^\top w}^{h_1(\theta,w,x)} \right) - \mathbb{E}_{(x,c)\sim P_n} \left( \min_{w\in W} \overbrace{\hat{c}_\theta(x)^\top w}^{h_2(\theta,w,x)} \right)
$$

$W$ is a convex bounded set, and for all $x \in \mathbb{R}^k$ and the two functions $w \in \mathbb{R}^d$, $\theta \longmapsto h_1(\theta,w,x)$, $\theta \longmapsto h_2(\theta,w,x)$ are both differentiable with respect to $\theta$. Moreover, $\frac{\partial h_1}{\partial \theta} \longmapsto \nabla\hat{c}_\theta(x)^\top w$ and $\frac{\partial h_2}{\partial \theta} \longmapsto \nabla\hat{c}_\theta(x)^\top w$ (where $\nabla\hat{c}_\theta(x)$ is the jacobian of $\theta \longmapsto \hat{c}_\theta(x)$ at $\theta$) are both continuous with respect to $w$ for all $\theta \in \mathbb{R}^m$ and $x \in \mathbb{R}^k$. Hence, using Danskin's theorem, we can say that for all $x \in \mathbb{R}^k$, $\overline{h}_1(\theta,x) = \min_{w\in W} h_1(\theta,w,x)$ and $\overline{h}_2(\theta,x) := \min_{w\in W} h_2(\theta,w,x)$ are both subdifferentiable, and that for all $\theta \in \mathbb{R}^m$, $\partial\overline{h}_1(\theta,x) = \mathrm{conv}\{\nabla\hat{c}_\theta(x)^\top w, \ w \in \arg\min_{w\in W} h_1(\theta,w,x)\}$, $\partial\overline{h}_2(\theta,x) = \mathrm{conv}\{\nabla\hat{c}_\theta(x)^\top w, \ w \in \arg\min_{w\in W} h_2(\theta,w,x)\}$. These two sets are bounded by $B_W B_\Phi$ because of Assumption 3. This means that the two functions in the equality above are both $B_W B_\Phi$ lipschitz, which makes $f_{\tilde{y},P_n}$ $2B_W B_\Phi$ lipschitz.

We can also easily prove that the inequality above is also true when we replace $P_n$ by $P$. Hence, we can deduce that

$$
\begin{aligned}
\left| \ell_{P_n}^\beta(\theta) - \ell_P^\beta(\theta) - \left( \ell_{P_n}^\beta(\theta') - \ell_P^\beta(\theta') \right) \right| &\leq \left| \ell_{P_n}^\beta(\theta) - \ell_{P_n}^\beta(\theta') \right| + \left| \ell_P^\beta(\theta) - \ell_P^\beta(\theta') \right| \\
&\leq |f_{\tilde{y},P_n}(\theta) - f_{\tilde{y},P_n}(\theta')| + |f_{\tilde{y},P}(\theta) - f_{\tilde{y},P}(\theta')| \\
&\leq 4B_W B_\Phi \|\theta - \theta'\|,
\end{aligned}
$$

which yields the desired result of step 1.

**Step 2.** We will now prove the desired generalization bound by bounding the generalization gap locally in balls covering the set which represents the possible values of $\theta^\star$. Let $\mathcal{B}$ be a set of balls of radius $\gamma > 0$ (for the norm $\|.\|$) such that $\bigcup_{B\in\mathcal{B}} B = B_{\|.\|}(0,D)$. According to (Wainwright, 2019), it is possible to choose $\mathcal{B}$ such that $\log|\mathcal{B}| \leq d\log\left(1 + \frac{2D}{\gamma}\right)$. For every $B \in \mathcal{B}$, let $\theta_B$ be an element of $B$. For a given $\varepsilon > 0$ we would like to bound the probability $\mathbb{P}\left( \left| \ell_{P_n}^\beta(\theta) - \ell_P^\beta(\theta) \right| > \varepsilon \right)$

using Hoeffding's inequality. In order to do that, we first notice that using Lagrangian duality, we get

$$
\begin{aligned}
\ell_P^\beta(\theta) - \ell_{P_n}^\beta(\theta) = &\max_{y \geq 0} \min_{\overline{w}_P^\beta \in W_P} \mathbb{E}_{(x,c) \sim P} \left( (\hat{c}_\theta(x) + yc(x))^\top \overline{w}_P^\beta(x) \right) - y\beta - \min_{w_P \in W_P} \mathbb{E}_{(x,c) \sim P} \left( \hat{c}_\theta(x)^\top w_P(x) \right) \\
&- \max_{y \geq 0} \min_{\overline{w}_{P_n}^\beta \in W_{P_n}} \mathbb{E}_{(x,c) \sim P_n} \left( (\hat{c}_\theta(x) + yc(x))^\top \overline{w}_{P_n}^\beta(x) \right) - y\beta - \min_{w_{P_n} \in W_{P_n}} \mathbb{E}_{(x,c) \sim P_n} \left( \hat{c}_\theta(x)^\top w_{P_n}(x) \right) \\
= &\min_{\overline{w}_P^\beta \in W_P} \mathbb{E}_{(x,c) \sim P} \left( (\hat{c}_\theta(x) + y_P^\star(\theta)c(x))^\top \overline{w}_P^\beta(x) \right) - y_P^\star(\theta)\beta - \min_{w_P \in W_P} \mathbb{E}_{(x,c) \sim P} \left( \hat{c}_\theta(x)^\top w_P(x) \right) \\
&- \left( \min_{\overline{w}_{P_n}^\beta \in W_{P_n}} \mathbb{E}_{(x,c) \sim P_n} \left( (\hat{c}_\theta(x) + y_{P_n}^\star(\theta)c(x))^\top \overline{w}_{P_n}^\beta(x) \right) - y_{P_n}^\star(\theta)\beta \right. \\
&\left. - \min_{w_{P_n} \in W_{P_n}} \mathbb{E}_{(x,c) \sim P_n} \left( \hat{c}_\theta(x)^\top w_{P_n}(x) \right) \right),
\end{aligned}
$$

where $y_P^\star(\theta)$ and $y_{P_n}^\star(\theta)$ are respectively optimal dual variables in the bottom and top line above. As we have seen before, replacing $y_{P_n}^\star(\theta)$ by $y_P^\star(\theta)$ makes the above term larger. Hence, we have

$$
\begin{aligned}
\ell_P^\beta(\theta) - \ell_{P_n}^\beta(\theta) \leq &\min_{\overline{w}_P^\beta \in W_P} \mathbb{E}_{(x,c) \sim P} \left( (\hat{c}_\theta(x) + y_P^\star(\theta)c(x))^\top \overline{w}_P^\beta(x) \right) - \min_{w_P \in W_P} \mathbb{E}_{(x,c) \sim P} \left( \hat{c}_\theta(x)^\top w_P(x) \right) \\
&- \left( \min_{\overline{w}_{P_n}^\beta \in W_{P_n}} \mathbb{E}_{(x,c) \sim P_n} \left( (\hat{c}_\theta(x) + y_P^\star(\theta)c(x))^\top \overline{w}_{P_n}^\beta(x) \right) - \min_{w_{P_n} \in W_{P_n}} \mathbb{E}_{(x,c) \sim P_n} \left( \hat{c}_\theta(x)^\top w_{P_n}(x) \right) \right) \\
= &\underbrace{\mathbb{E}_{(x,c) \sim P} \left( \min_{w \in W} (\hat{c}_\theta(x) + y_P^\star(\theta)c(x))^\top w - \min_{w \in W} \hat{c}_\theta(x)^\top w \right)}_{\tilde{\ell}_P^\beta(\theta)} \\
&- \underbrace{\mathbb{E}_{(x,c) \sim P_n} \left( \min_{w \in W} (\hat{c}_\theta(x) + y_P^\star(\theta)c(x))^\top w - \min_{w \in W} \hat{c}_\theta(x)^\top w \right)}_{\tilde{\ell}_{P_n}^\beta(\theta)}
\end{aligned}
$$

Hence, we have $\mathbb{P} \left( \ell_P^\beta(\theta) - \ell_{P_n}^\beta(\theta) > \varepsilon \right) \leq \mathbb{P} \left( \tilde{\ell}_P^\beta(\theta) - \tilde{\ell}_{P_n}^\beta(\theta) > \varepsilon \right)$. This enables us to apply Hoeffding's inequality to the term on the right given that we bound the random variable

$$
m(x, \theta) := \min_{w \in W} (\hat{c}_\theta(x) + y_P^\star(\theta)c(x))^\top w - \min_{w \in W} \hat{c}_\theta(x)^\top w.
$$

It is easy to see that using Cauchy-Schwartz inequality, we have

$$
|m(x, \theta)| \leq (\|\hat{c}_\theta(x)\| + |y_P^\star(\theta)| \|c(x)\|) B_W + \|\hat{c}_\theta(x)\| B_W
$$

In order to upper bound the right-hand side, we need to upper bound $y_P^\star(\theta)$. In order to do this, we prove that the Lagrange multiplier $y_P^\star(\theta)$ is bounded for any $\theta \in \mathbb{R}^m$. Indeed, we have

$$
\min_{w_P \in W_P} \mathbb{E}_{(x,c) \sim P}(\hat{c}_\theta(x)^\top w_P(x)) \leq \min_{w_P \in W_P} \max_{y \geq 0} \mathbb{E}_{(x,c) \sim P}(\hat{c}_\theta(x)^\top w_P(x)) + y(\mathbb{E}_{(x,c) \sim P}(c(x)^\top w_P(x)) - \beta) \tag{24}
$$

$$
= \min_{w_P \in W_P} \mathbb{E}_{(x,c) \sim P}(\hat{c}_\theta(x)^\top w_P(x)) + y_P^\star(\theta)(\mathbb{E}_{(x,c) \sim P}(c(x)^\top w_P(x)) - \beta) \tag{25}
$$

$$
\leq \mathbb{E}_{(x,c) \sim P}(\hat{c}_\theta(x)^\top w(c(x))) + y_P^\star(\theta)(\mathbb{E}_{(x,c) \sim P}(c(x)^\top w(c(x))) - \beta). \tag{26}
$$

Inequality (24) holds because the left-hand side is the evaluation of the right-hand side at $y = 0$. Hence,

$$
\min_{w_P \in W_P} \mathbb{E}_{(x,c) \sim P}(\hat{c}_\theta(x)^\top w_P(x)) \leq \mathbb{E}_{(x,c) \sim P}(\hat{c}_\theta(x)^\top w(c(x))) + y_P^\star(\theta)(\mathbb{E}_{(x,c) \sim P}(c(x)^\top w(c(x))) - \beta)
$$

$$
= \mathbb{E}_{(x,c) \sim P}(\hat{c}_\theta(x)^\top w(c(x))) + y_P^\star(\theta)(\beta_{\min,P} - \beta).
$$

This yields

$$
y_P^\star(\theta) \leq \frac{\mathbb{E}_{(x,c) \sim P}(\hat{c}_\theta(x)^\top w(c(x))) - \mathbb{E}_{(x,c) \sim P}(\hat{c}_\theta(x)^\top w(\hat{c}_\theta(x)))}{\beta - \beta_{\min,P}} \leq \frac{2DB_W B_\Phi}{\beta - \beta_{\min,P}}.
$$

Hence, we get the following upper bound for the random variables we are working with

$$|m(x,\theta)| \leq 2DB_W B_\Phi + KB_W \frac{2DB_W B_\Phi}{\beta - \beta_{\min,P}} := U$$

we can apply Hoeffding's inequality, and say that

$$\mathbb{P}\left(\tilde{\ell}_P^\beta(\theta) - \tilde{\ell}_{P_n}^\beta(\theta) > \varepsilon\right) \leq 2e^{-\frac{n\varepsilon^2}{2U^2}}.$$

This yields

$$\mathbb{P}\left(\sup_{B \in \mathcal{B}} \left(\tilde{\ell}_P^\beta(\theta_B) - \tilde{\ell}_{P_n}^\beta(\theta_B)\right) > \varepsilon\right) \leq |\mathcal{B}|\, 2e^{-\frac{n\varepsilon^2}{2U^2}}.$$

By denoting $\delta := |\mathcal{B}|\, 2e^{-\frac{n\varepsilon^2}{2U^2}}$, we can say that with probability at least $1 - \delta$, we have

$$\forall B \in \mathcal{B},\ \tilde{\ell}_P^\beta(\theta_B) - \tilde{\ell}_{P_n}^\beta(\theta_B) \leq U\sqrt{2}\sqrt{\frac{\log|\mathcal{B}| + \log\frac{2}{\delta}}{n}},$$

which gives

$$\forall B \in \mathcal{B},\ \ell_P^\beta(\theta_B) - \ell_{P_n}^\beta(\theta_B) \leq U\sqrt{2}\sqrt{\frac{\log|\mathcal{B}| + \log\frac{2}{\delta}}{n}} \leq U\sqrt{2}\sqrt{\frac{d\log\left(1 + \frac{2D}{\gamma}\right) + \log\frac{2}{\delta}}{n}}.$$

Now we have for every $\theta \in B_{\|.\|}(0, D)$, denoting $B \in \mathcal{B}$ such that $\theta \in B$,

$$\ell_P^\beta(\theta) - \ell_{P_n}^\beta(\theta) \leq \ell_P^\beta(\theta_B) - \ell_{P_n}^\beta(\theta_B) + \left|\ell_{P_n}^\beta(\theta) - \ell_P^\beta(\theta) - \left(\ell_{P_n}^\beta(\theta_B) - \ell_P^\beta(\theta_B)\right)\right| \tag{27}$$

$$\leq U\sqrt{2}\sqrt{\frac{d\log\left(1 + \frac{2D}{\gamma}\right) + \log\frac{2}{\delta}}{n}} + 4B_W B_\Phi \|\theta - \theta_B\| \tag{28}$$

$$\leq U\sqrt{2}\sqrt{\frac{d\log\left(1 + \frac{2D}{\gamma}\right) + \log\frac{2}{\delta}}{n}} + 4B_W B_\Phi \gamma. \tag{29}$$

Inequality (29) is resulting from the Lipschitz inequality (19). Taking $\gamma = O\left(\sqrt{\frac{\log\frac{1}{\delta}}{n}}\right)$, we get

$$\ell_P^\beta(\theta) - \ell_{P_n}^\beta(\theta) \leq O\left((U + 4B_W B_\Phi)\sqrt{\frac{\log\frac{1}{\delta}}{n}}\right).$$

Since we have $\ell_{P_n}^\beta(\theta^\star) \leq \varepsilon$, we can replace in the inequality above $\theta$ by $\theta^\star$ and get

$$\ell_P^\beta(\theta^\star) \leq \ell_{P_n}^\beta(\theta^\star) + O\left((U + 4B_W B_\Phi)\sqrt{\frac{\log\frac{1}{\delta}}{n}}\right) \leq \varepsilon + O\left(\frac{1}{\beta - \beta_{\min,P}}\sqrt{\frac{\log\frac{1}{\delta}}{n}}\right).$$

$\square$

## A.10. Proof of Lemma 1

***Proof.*** Since the only difference between the right and left optimization problems in $\ell_P^\beta$ is an additional constraint added to the right problem, it is clear that it will yield a higher value than the right one for any $\theta \in \mathbb{R}^m$, and consequently we have that indeed $\ell_P^\beta$ is positive. $\square$

### A.11. Proof of Proposition 1

***Proof.*** We denote for every $w \in W$ and $\theta \in \mathbb{R}^m$, $f(\theta; w) = -\hat{c}_\theta(x)^\top w$. We want to prove that for every $w \in W$, $\theta \longrightarrow f(\theta; w)$ is weakly convex. More precisely, we want to prove that there exists $\alpha \geq 0$ such that for every $w \in W$, $\theta \longmapsto f(\theta; w) + \alpha \|\theta\|^2$ is convex. We have for every $\theta_1, \theta_2 \in \mathbb{R}^m$, using assumptions 6 and 2,

$$\begin{aligned}
\|\nabla f(\theta_1; w) - \nabla f(\theta_2; w)\| &= \left\| (\nabla \hat{c}_{\theta_1}(x) - \nabla \hat{c}_{\theta_2}(x))^\top w \right\| \\
&\leq B_W \|\nabla \hat{c}_{\theta_1}(x) - \nabla \hat{c}_{\theta_2}(x)\| \\
&\leq B_W B_L \|\theta_1 - \theta_2\|.
\end{aligned}$$

We denote for every $\theta \in \mathbb{R}^m$ and $w \in W$, $h(\theta; w) = f(\theta; w) + 2 B_W B_L \|\theta\|^2$. We have for every $\theta_1, \theta_2 \in \mathbb{R}^m$,

$$\begin{aligned}
(\nabla h(\theta_1; w) - \nabla h(\theta_2; w))^\top (\theta_1 - \theta_2) &= (\nabla f(\theta_1; w) - \nabla f(\theta_2; w))^\top (\theta_1 - \theta_2) + 2 B_W B_L \|x - y\|^2 \\
&\geq -\|\nabla f(\theta_1; w) - \nabla f(\theta_2; w)\| \|\theta_1 - \theta_2\| + 2 B_W B_L \|\theta_1 - \theta_2\|^2 \\
&\geq -B_W B_L \|\theta_1 - \theta_2\|^2 + 2 B_W B_L \|\theta_1 - \theta_2\|^2 \\
&= B_W B_L \|\theta_1 - \theta_2\|^2 \geq 0.
\end{aligned}$$

Hence, for every $w \in W$, $\theta \longmapsto h(\theta; w)$ is convex. This implies that for any measurable mapping $w : \mathbb{R}^k \longrightarrow W$, $\theta \longmapsto \mathbb{E}_{(x,c) \sim P_n}(h(\theta; w(x)))$ is convex. Consequently, $\theta \longmapsto \max_{w_{P_n} \in W_{P_n}} \mathbb{E}_{(x,c) \sim P_n}(h(\theta; w(x)))$ and $\theta \longmapsto \max_{w_{P_n} \in \overline{W}_{P_n}^\beta} \mathbb{E}_{(x,c) \sim P_n}(h(\theta; w(x)))$ are both maximums of a family of convex functions, and hence are also convex functions. Finally we have for every $\theta \in \mathbb{R}^m$,

$$\begin{aligned}
\max_{w_{P_n} \in W_{P_n}} \mathbb{E}_{(x,c) \sim P_n}(h(\theta; w(x))) &= g_{P_n}(\theta) + 2 B_L \|\theta\|^2 \\
\max_{w_{P_n} \in \overline{W}_{P_n}^\beta} \mathbb{E}_{(x,c) \sim P_n}(h(\theta; w(x))) &= \overline{g}_{P_n}^\beta(\theta) + 2 B_L \|\theta\|^2.
\end{aligned}$$

Finally, we can write for every $\theta \in \mathbb{R}^m$,

$$\ell_{P_n}^\beta(\theta) = g_{P_n}(\theta) + 2 B_L \|\theta\|^2 - \left( \overline{g}_{P_n}^\beta(\theta) + 2 B_L \|\theta\|^2 \right)$$

which is indeed a difference of convex functions. $\qquad \square$

### A.12. Proof of Theorem 3

***Proof.*** Let $\theta \in \mathbb{R}^m$ and $t > 0$. Assumption 5 gives

$$\mathbb{P}(0 < \Delta_\theta(x) \leq \|\theta\| t) \leq \left( \frac{\gamma t}{B} \right)^\alpha.$$

where

$$\Delta_\theta(x) = \begin{cases} \min_{w \in W^\angle \setminus W^\star(\hat{c}_\theta(x))} \hat{c}_\theta(x)^\top w - \min_{w \in W^\angle} \hat{c}_\theta(x)^\top w & \text{if } W^\star(\hat{c}_\theta(x)) \neq W \\ 0 & \text{else.} \end{cases}$$

We have for $\theta \neq 0$, denoting $W^{\angle} = \{w_1, \ldots, w_N\}$, $N \in \mathbb{N}$,

$$\Delta_\theta(x) > \|\theta\| \, t \implies \forall w_{\angle} \in W^{\angle} \setminus \{w(\hat{c}_\theta)\}, \; \hat{c}_\theta(x)^\top w_{\angle} - \hat{c}_\theta(x)^\top w(\hat{c}_\theta(x)) > \|\theta\| \, t$$

$$\implies \forall w_{\angle} \in W^{\angle} \setminus \{w(\hat{c}_\theta)\}, \; \hat{c}_\theta(x)^\top w_{\angle} - \hat{c}_\theta(x)^\top w(\hat{c}_\theta(x)) > \|\theta\| \frac{\|w(\hat{c}_\theta(x)) - w_{\angle}\|}{2B_W} t$$

$$\implies \forall \lambda_1, \ldots, \lambda_n \geq 0 \text{ s.t. } \lambda_1 + \cdots + \lambda_n = 1,$$

$$\sum_{i=1}^N \lambda_i \left( \hat{c}_\theta(x)^\top w_i - \hat{c}_\theta(x)^\top w(\hat{c}_\theta(x)) \right) > \frac{1}{2B_W} \|\theta\| \, t \sum_{i=1}^N \lambda_i \|w(\hat{c}_\theta(x)) - w_i\|$$

$$\implies \forall \lambda_1, \ldots, \lambda_n \geq 0 \text{ s.t. } \lambda_1 + \cdots + \lambda_n = 1,$$

$$\hat{c}_\theta(x)^\top \sum_{i=1}^N \lambda_i w_i - \hat{c}_\theta(x)^\top w(\hat{c}_\theta(x)) > \frac{1}{2B_W} \|\theta\| \, t \left\| w(\hat{c}_\theta(x)) - \sum_{i=1}^N \lambda_i w_i \right\|$$

$$\implies \forall w \in W, \; \hat{c}_\theta(x)^\top w - \hat{c}_\theta(x)^\top w(\hat{c}_\theta(x)) > \frac{1}{2B_W} \|\theta\| \, \|w(\hat{c}_\theta(x)) - w\| \, t.$$

Hence, we have

$$\mathbb{P}\left( \forall w \in W, \; \hat{c}_\theta(x)^\top w - \hat{c}_\theta(x)^\top w(\hat{c}_\theta(x)) > \frac{1}{2B_W} \|\theta\| \, \|w(\hat{c}_\theta(x)) - w\| \, t \right) \geq \mathbb{P}\left( \Delta_\theta(x) > \|\theta\| \, t \right) > 1 - \left( \frac{\gamma t}{B} \right)^\alpha \quad (30)$$

We denote for $w \in W$, $t \geq 0$, and $\theta \in \mathbb{R}^m \setminus \{0\}$,

$$G(x, \theta, w, t) = \hat{c}_\theta(x)^\top w - \hat{c}_\theta(x)^\top w(\hat{c}_\theta(x)) - \frac{1}{2B_W} \|\theta\| \, \|w(\hat{c}_\theta(x)) - w\| \, t$$

and the event $A(x, \theta, t)$ defined by

$$A(x, \theta, t) = \{ \forall w \in W, \; G(x, \theta, w(x), t) > 0 \} .$$

Hence, we have for every mapping $w : \mathbb{R}^k \longrightarrow w$ and $\theta \in \mathbb{R}^m \setminus \{0\}$,

$$\mathbb{E}_{(x,c) \sim P}(G(x, \theta, w(x), t)) = \mathbb{E}_{(x,c) \sim P}(G(x, \theta, w(x), t) 1_{A(x,\theta,t)}) + \mathbb{E}_{(x,c) \sim P}(G(x, \theta, w(x), t) 1_{A(x,\theta,t)^c})$$
$$\geq \mathbb{E}_{(x,c) \sim P}(G(x, \theta, w(x), t) 1_{A(x,\theta,t)^c}).$$

Furthermore, we have for every $w \in W$

$$|G(x, \theta, w, t)| = \left| \hat{c}_\theta(x)^\top w - \hat{c}_\theta(x)^\top w(\hat{c}_\theta(x)) - \frac{1}{2B_W} \|\theta\| \, \|w(\hat{c}_\theta(x)) - w\| \, t \right|$$
$$\leq B_\Phi B_W \|\theta\| + B_\Phi B_W \|\theta\| + \|\theta\| \, t = \|\theta\| \, (2B_W B_\Phi + t) .$$

Notice that (30) can be rewritten as $\mathbb{P}(A(x, \theta, t)) > 1 - \left( \frac{\gamma t}{B} \right)^\alpha$. Hence, we have

$$\mathbb{E}_{(x,c) \sim P}(G(x, \theta, w(x), t)) \geq -(2B_W B_\Phi + t) \|\theta\| \, \mathbb{P}(A(x, \theta, t)^c)$$
$$\geq -(2B_W B_\Phi + t) \|\theta\| \left( \frac{\gamma t}{B} \right)^\alpha .$$

In conclusion, we have for every $t \geq 0$ and every $\theta \in \mathbb{R}^m$, and mapping $w : \mathbb{R}^k \longrightarrow W$,

$$\mathbb{E}_{(x,c) \sim P}\left( \hat{c}_\theta(x)^\top w(x) - \hat{c}_\theta(x)^\top w(\hat{c}_\theta(x)) \right) \geq \frac{1}{2B_W} \|\theta\| \, \mathbb{E}_{(x,c) \sim P}(\|w(\hat{c}_\theta(x)) - w(x)\|) \, t - (2B_W B_\Phi + t) \|\theta\| \left( \frac{\gamma t}{B} \right)^\alpha .$$

the inequality above is trivially verified when $\theta = 0$. Also, when taking $w \in \arg\min_{w \in \overline{W}_P^\beta} \mathbb{E}_{(x,c) \sim P}\left( \hat{c}_\theta(x)^\top w(x) \right)$, we get

$$\ell_P^\beta(\theta) \geq \frac{1}{2B_W} \|\theta\| \, \mathbb{E}_{(x,c) \sim P}(\|w(\hat{c}_\theta(x)) - w(x)\|) \, t - (2B_W B_\Phi + t) \|\theta\| \left( \frac{\gamma t}{B} \right)^\alpha$$
$$\geq \frac{\|\theta\| \, t}{2K B_W} (\ell_P(\theta) - \beta) - (2B_W B_\Phi + t) \|\theta\| \left( \frac{\gamma t}{B} \right)^\alpha$$

Let $\varepsilon > 0$. Assume that $\theta$ is bounded away from 0. We have

$$\ell_P(\theta) - \beta \leq \frac{2KB_W}{t} \left( \frac{\ell_P^\beta(\theta)}{\|\theta\|} + (2B_W B_\Phi + t) \left( \frac{\gamma t}{B} \right)^\alpha \right) \leq O\left( \frac{1}{t} \left( \ell_P^\beta(\theta) + t^\alpha \right) \right)$$

To optimize the right hand side and ignoring the constant, we take $t = \ell_P^\beta(\theta)^{\frac{1}{1+\alpha}}$. This gives

$$\ell_P(\theta) \leq \beta + O\left( \ell_P^\beta(\theta)^{1 - \frac{1}{1+\alpha}} \right).$$

$\square$

### A.13. A more complete version of Theorem 4

**Theorem 5.**

1. *For all $\theta \in \mathbb{R}^m$, $\overline{g}_{P_n}^\beta$ and $g_{P_n}$ have a non-empty subgradient at $\theta$, and $\ell_{P_n}^\beta(\theta) = \theta^\top v(\theta)$ for some $v(\theta) \in \partial \ell_{P_n}^\beta := \partial \overline{g}_{P_n}^\beta(\theta) - \partial g_{P_n}^\beta(\theta) := \{u_\beta - u, \ (u_\beta, u) \in \partial \overline{g}_{P_n}^\beta(\theta) \times \partial g_{P_n}(\theta)\}$, where $\partial \overline{g}_{P_n}^\beta(\theta)$ and $\partial g_{P_n}(\theta)$ are respectively the subgradients of $\overline{g}_{P_n}^\beta$ and $g_{P_n}$ at $\theta$;*

2. *For all $\theta \in \mathbb{R}^m$, if $v(\theta) = 0$ for some $v(\theta) \in \partial \ell_{P_n}^\beta(\theta)$, then $\ell_{P_n}^\beta(\theta) = 0$, hence $\theta$ is a minimizer.*

3. *$r_{P_n}^\beta$ is everywhere differentiable. If $\lambda$ is a stationary point of $r_{P_n}^\beta$ such that $\hat{c}_{\overline{\theta}_{P_n}^\beta(\lambda)}(x) \neq 0$ almost surely, then $\overline{\theta}_{P_n}^\beta(\lambda)$ is a global minimum of $\ell_{P_n}^\beta$. In particular, under assumptions 2 and 3, we have for every $\theta \in \mathbb{R}^m$ and $\varepsilon \geq 0$,*

$$\left\| \nabla r_{P_n}^\beta \right\| \leq \varepsilon \Longrightarrow \ell_{P_n}^\beta(\theta) \leq 8B_W B_\Phi \varepsilon.$$

*Proof.*

1. We denote for all $(\theta, v) \in \mathbb{R}^m \times V_P$, $\phi(\theta, v) = \theta^\top v$, $v_\beta$ and for all $(\theta, v_\beta) \in \mathbb{R}^m \times V_P$, $\phi(\theta, v_\beta) = \theta^\top v_\beta$, In this case, for all $\theta \in \mathbb{R}^m$, the loss writes as

$$\ell_P^\beta(\theta) = g_P(\theta) - \overline{g}_P^\beta(\theta) = \min_{v_\beta \in V_P^\beta} \phi(\theta, v_\beta) - \min_{v \in V_P} \phi(\theta, v).$$

Let $\mathcal{V}_P^\beta(\theta)$ and $\mathcal{V}_P(\theta)$ be the set of minimizers of respectively $v_\beta \longmapsto \phi(\theta, v_\beta)$ over $V_P^\beta$ and $v \longmapsto \phi(\theta, v)$ over $V_P$. Given that $V_P$ and $V_P^\beta$ are compact sets, and $\theta \longmapsto \phi(\theta, v)$ for all $v \in V_P \cup V_P^\beta$ is differentiable, and $\phi$ is continuous, we can say by Danskin's theorem that for all $\theta \in \mathbb{R}^m$,

$$\partial g_P(\theta) = \text{conv}\left\{ \frac{\partial \phi(\theta, v)}{\partial \theta}, \ v \in \mathcal{V}_P(\theta) \right\} = \text{conv}\, \mathcal{V}_P(\theta) \underset{(*)}{=} \mathcal{V}_P(\theta),$$

$$\partial \overline{g}_P^\beta(\theta) = \text{conv}\left\{ \frac{\partial \phi(\theta, v_\beta)}{\partial \theta}, \ v_\beta \in \mathcal{V}_P^\beta(\theta) \right\} = \text{conv}\, \mathcal{V}_P^\beta(\theta) \underset{(**)}{=} \mathcal{V}_P^\beta(\theta).$$

The two inequalities $(*)$ and $(**)$ are due to the fact that $\mathcal{V}_P(\theta)$ and $\mathcal{V}_P^\beta(\theta)$ are convex sets. Furthermore, we have for all $\theta \in \mathbb{R}^m$,

$$\ell_P^\beta(\theta) = \min_{v_\beta \in V_P^\beta} \phi(\theta, v_\beta) - \min_{v \in V_P} \phi(\theta, v) = \theta^\top v_\beta^\star - \theta^\top v^\star = \theta^\top \underbrace{(v_\beta^\star - v^\star)}_{:= v(\theta)},$$

where $(v_\beta^\star, v^\star) \in \mathcal{V}_P^\beta(\theta) \times \mathcal{V}_P(\theta)$. All of the above clearly yields

$$v(\theta) \in \mathcal{V}_P^\beta(\theta) - \mathcal{V}_P(\theta) = \partial \overline{g}_P^\beta(\theta) - \partial g_P(\theta),$$

which is the result we were seeking to prove.

2. From the above, it is clear that for a given $\theta \in \mathbb{R}^m$ if $v(\theta) = 0$, then $\ell_P^\beta(\theta) = 0$. Furthermore, using the very first definition of $\ell_P^\beta$, since $\overline{W}_P^\beta \subset W_P$, we have for all $\theta \in \mathbb{R}^m$,

$$\ell_P^\beta(\theta) = \min_{w_P \in \overline{W}_P^\beta} \mathbb{E}_{(c,x) \sim P}\left((\hat{c}_\theta(x)^\top w_P(x))\right) - \min_{w_P \in W_P} \mathbb{E}_{(c,x) \sim P}\left(\hat{c}_\theta(x)^\top w_P(x)\right) \geq 0.$$

In conclusion, 0 is a lower bound of $\ell_P^\beta$, and if for a given $\theta$, $v(\theta) = 0$, then $\ell_P^\beta(\theta) = 0$, i.e. $\theta$ is a minimizer of $\ell_P^\beta$.

3. $r_P^\beta$ is the difference of the Moreau envelope of two convex functions, hence it is the difference between two differentiable convex functions. this yields that $r_P^\beta$ is differentiable. Furthermore, if $\lambda$ is a stationary point of $r_P^\beta$ then using proposition 1 from (Sun and Sun, 2021), $\theta_P^\beta(\lambda)$ is indeed a stationary point of $\ell_P^\beta$. Furthermore, we have for every $\varepsilon \geq 0$, We consider $\lambda \in \mathbb{R}^m$ an $\varepsilon$-stationary point of $r_{P_n}^\beta$, i.e. $\left\|\nabla r_{P_n}^\beta(\lambda)\right\| \leq \varepsilon$. Using Danskin's theorem, we get

$$\nabla r_{P_n}^\beta(\lambda) = \nabla M_{P_n}(\lambda) - \nabla \overline{M}_{P_n}^\beta(\lambda)$$
$$= \lambda - \theta_{P_n}(\lambda) - (\lambda - \overline{\theta}_{P_n}^\beta(\lambda))$$
$$= \overline{\theta}_{P_n}^\beta(\lambda) - \theta_{P_n}(\lambda),$$

Where $\overline{\theta}_{P_n}^\beta(\lambda) = \arg\min_{\theta \in \mathbb{R}^m} g_{P_n}^\beta(\theta) + \frac{1}{2}\|\lambda - \theta\|^2$ and $\theta_{P_n}(\lambda) = \arg\min_{\theta \in \mathbb{R}^m} g_{P_n}(\theta) + \frac{1}{2}\|\lambda - \theta\|^2$. By looking at the proof of Lemma 2, we can easily see that we also have $\theta_{P_n}(\lambda) = \lambda + v_{P_n}(\theta_{P_n}(\lambda))$ where $v_{P_n}(\theta_{P_n}(\lambda)) \in \arg\min_{v \in V_{P_n}} \theta_{P_n}(\lambda)^\top v = \partial g_{P_n}(\theta_{P_n}(\lambda))$ and $\overline{\theta}_{P_n}^\beta(\lambda) = \lambda + v_{P_n}^\beta(\overline{\theta}_{P_n}^\beta(\lambda))$ where $v_{P_n}^\beta(\overline{\theta}_{P_n}^\beta(\lambda)) \in \arg\min_{v \in \overline{V}_{P_n}^\beta} \overline{\theta}_{P_n}^\beta(\lambda)^\top v = \partial g_{P_n}^\beta(\overline{\theta}_{P_n}^\beta(\lambda))$. Let $v_{P_n}(\overline{\theta}_{P_n}^\beta(\lambda)) \in \arg\min_{v \in V_{P_n}} \overline{\theta}_{P_n}^\beta(\lambda)^\top v = \partial g_{P_n}(\overline{\theta}_{P_n}^\beta(\lambda))$. We have

$$\ell_{P_n}^\beta(\overline{\theta}_{P_n}^\beta(\lambda)) = \overline{\theta}_{P_n}^\beta(\lambda)^\top v_{P_n}^\beta(\overline{\theta}_{P_n}^\beta(\lambda)) - \overline{\theta}_{P_n}^\beta(\lambda) v_{P_n}(\overline{\theta}_{P_n}^\beta(\lambda)) \tag{31}$$

$$\leq \left|\overline{\theta}_{P_n}^\beta(\lambda)^\top v_{P_n}^\beta(\overline{\theta}_{P_n}^\beta(\lambda)) - \overline{\theta}_{P_n}^\beta(\lambda)^\top v_{P_n}(\theta_{P_n}(\lambda))\right| \tag{32}$$

$$+ \left|\theta_{P_n}(\lambda)^\top v_{P_n}(\theta_{P_n}(\lambda)) - \overline{\theta}_{P_n}^\beta(\lambda)^\top v_{P_n}(\overline{\theta}_{P_n}^\beta(\lambda))\right| \tag{33}$$

$$+ \left|\theta_{P_n}(\lambda)^\top v_{P_n}(\theta_{P_n}(\lambda)) - \overline{\theta}_{P_n}^\beta(\lambda)^\top v_{P_n}(\theta_{P_n}(\lambda))\right| \tag{34}$$

$$\leq \left\|\overline{\theta}_{P_n}^\beta(\lambda)\right\| \left\|v_{P_n}^\beta(\overline{\theta}_{P_n}^\beta(\lambda)) - v_{P_n}(\theta_{P_n}(\lambda))\right\| + \left|\kappa(\theta_{P_n}^\beta(\lambda)) - \kappa(\theta_{P_n}(\lambda))\right| \tag{35}$$

$$+ \|v_{P_n}(\theta_{P_n}(\lambda))\| \left\|\theta_{P_n}(\lambda) - \overline{\theta}_{P_n}^\beta(\lambda)\right\| \tag{36}$$

$$\leq (2B_W B_\Phi + 5B_W B_\Phi)\varepsilon + \left|\kappa(\theta_{P_n}^\beta(\lambda)) - \kappa(\theta_{P_n}(\lambda))\right| \tag{37}$$

$$\leq (3B_W B_\Phi + 5B_W B_\Phi)\varepsilon = 8B_W B_\Phi \varepsilon \tag{38}$$

Here, $\kappa$ is defined for any $\theta \in \mathbb{R}^m$ as $\kappa(\theta) = \min_{v \in V_{P_n}} \theta^\top v$. The last equality 38 is due to the fact that the subgradient of $\kappa$ at some $\theta \in \mathbb{R}^m$ is in $V_{P_n}$ (which can be proven thanks to Danskin's theorem) and is hence bounded by $B_W B_\Phi$. We can see that inequality 38 is indeed the inequality we were seeking to obtain. Taking $\varepsilon = 0$ gives that stationarity for $r_{P_n}^\beta$ implies global optimality for $\ell_{P_n}^\beta$.

$\square$

## A.14. Key lemma to prove Proposition 2

**Lemma 2.** *For every $\lambda \in \mathbb{R}^m$ and $\beta \geq \beta_{\mathcal{H},P}^\star$, we denote $V_{P_n}^1 = V_{P_n}$ and $V_{P_n}^2 = \overline{V}_{P_n}^\beta$. We have for $i \in \{1, 2\}$,*
$\theta_{P_n}^i(\lambda) = \lambda + v_i$ *with* $v_i = \arg\min_{v \in V_{P_n}^i} \frac{1}{2}\|\lambda + v\|^2$ *and* $M_{P_n}^i(\lambda) = \frac{1}{2}\|\lambda\|^2 - \min_{v \in V_{P_n}^i} \frac{1}{2}\|\lambda + v\|^2$.

**Proof.** The main idea of the proof is to switch min and max in the definitions of $M_P^\beta$ and $\overline{M}_P^\beta$.

1. We have for all $\lambda \in \mathbb{R}^m$,

$$M_P(\lambda) = \min_{\theta \in \mathbb{R}^m} g_P^\beta(\theta) + \frac{1}{2} \|\lambda - \theta\|^2 \tag{39}$$

$$= \min_{\theta \in \mathbb{R}^m} \left( - \min_{w_P \in W_P} \mathbb{E}(\hat{c}_\theta(x)^\top w_P(x)) \right) + \frac{1}{2} \|\lambda - \theta\|^2 \tag{40}$$

$$= \min_{\theta \in \mathbb{R}^m} \max_{w_P \in W_P} -\mathbb{E}(\hat{c}_\theta(x)^\top w_P(x)) + \frac{1}{2} \|\lambda - \theta\|^2 \tag{41}$$

$$= \min_{\theta \in \mathbb{R}^m} \max_{w_P \in W_P} -\theta^\top \underbrace{\mathbb{E}(\Phi(x)^\top w_P(x))}_{v} + \frac{1}{2} \|\lambda - \theta\|^2 \tag{42}$$

$$= \min_{\theta \in \mathbb{R}^m} \max_{v \in V_P} -\theta^\top v + \frac{1}{2} \|\lambda - \theta\|^2 \tag{43}$$

$$= \max_{v \in V_P} \min_{\theta \in \mathbb{R}^m} -\theta^\top v + \frac{1}{2} \|\lambda - \theta\|^2 \tag{44}$$

The equality between 44 and 43 holds because of Sion's minimax theorem: since $\mathbb{R}^m$ and $V_P$ are convex sets, $v \longmapsto -\theta^\top v$ is upper semi-continuous and concave for any $\theta \in \mathbb{R}^m$, and $\theta \longmapsto -\theta^\top v + \frac{1}{2} \|\lambda - \theta\|^2$ is lower semicontinuous and convex for any $v \in V_P$, we can switch min and max. In 44, the minimum is reached when the gradient with respect to $\theta$ is zero, i.e. $-v + \theta - \lambda = 0$, which gives $\theta = v + \lambda$. In this case, we get

$$M_P(\lambda) = \max_{v \in V_P} -(v + \lambda)^\top v + \frac{1}{2} \|v\|^2$$

$$= \max_{v \in V_P} -\lambda^\top v - \frac{1}{2} \|v\|^2$$

$$= \frac{1}{2} \|\lambda\|^2 + \max_{v \in V_P} -\frac{1}{2} \|\lambda + v\|^2$$

$$= \frac{1}{2} \|\lambda\|^2 - \min_{v \in V_P} \frac{1}{2} \|\lambda + v\|^2.$$

The calculations above also give us immediately that $\theta_P(\lambda) = \lambda + v$ where $v = \arg\min_{v \in V_P} \frac{1}{2} \|\lambda + v\|^2$.

2. The proof of the second property is almost identical to the previous one. It suffices to replace $V_P$ by $\overline{V}_P^\beta$ and $W_P$ by $\overline{W}_P^\beta$.

$\square$

## A.15. Proof of Proposition 2

This proposition immediately follows from Lemma 2.

## A.16. More details on avoiding $\theta = 0$

**Theorem 6.** *We define $F_{V_{P_n}}(\lambda) := \arg\min_{\lambda' \in -V_{P_n}} \|\lambda - \lambda'\|$ and $F_{\overline{V}_{P_n}^\beta}(\lambda) := \arg\min_{\lambda' \in -\overline{V}_{P_n}^\beta} \|\lambda - \lambda'\|$ to be respectively the projection of $\lambda$ on $-V_{P_n}$ and on $-\overline{V}_{P_n}^\beta$. Suppose that $\lambda$ is an $\varepsilon$-solution of $f_{P_n}^\beta$, i.e. $\left\| \nabla f_{P_n}^\beta(\lambda) \right\| \leq \varepsilon$ for some $\varepsilon > 0$, and $\lambda \notin V_{P_n}^\beta$. We have $\left\| \nabla r_{P_n}^\beta \right\| \leq 27 B_W^2 B_\Phi^2 \varepsilon$. Furthermore, if $d(\lambda, -V_{P_n}) \leq B_W B_\Phi$, letting $\tilde{\lambda} = \lambda + r \frac{\lambda - F_{V_{P_n}}(\lambda)}{\|\lambda - F_{V_{P_n}}(\lambda)\|}$ with $r \in [3 B_W B_\Phi, 8 B_W B_\Phi]$ and $\varepsilon' = 27 B_W^2 B_\Phi^2 \varepsilon$, we have*

- *The value of $r_{P_n}^\beta(\tilde{\lambda})$ is dominated by $\varepsilon$. In particular, we have $r_{P_n}^\beta(\tilde{\lambda}) \leq 11 B_W B_\Phi \varepsilon'$ and $\left\| \nabla r_{P_n}^\beta(\tilde{\lambda}) \right\| \leq \sqrt{11 B_W B_\Phi \varepsilon'}$.*

- *The norm of the resulting candidate solution for $\ell_{P_n}^\beta$ is bounded from above and below. In particular, $B_W B_\Phi \leq \|\bar{\theta}_{P_n}^\beta(\tilde{\lambda})\| \leq 11 B_W B_\Phi$.*

Figure 5 summarizes the construction of $\tilde{\lambda}$.

The theorem above provides us with an algorithm to optimize $\ell_{P_n}^\beta$:

1. Run gradient descent on $r_{P_n}^\beta$.

2. If we obtain a solution reasonably far from $V_{P_n}$, we simply return that iterate as an output. Else, we run gradient descent on $f_{P_n}^\beta$.

3. If the gradient descent iterates do not converge to a common boundary to $-V_{P_n}$ and $-\overline{V}_{P_n}^\beta$, we return the final iterate as an output. Else, we use the procedure in Theorem 6 to obtain a near stationary point which is far from the boundary, and return it as an output.

***Proof.*** We will sequentially prove the two points. We assume that $\lambda \in \mathbb{R}^m$ is a stationary point of $f_{P_n}^\beta$. In this case, we have

$$0 = \nabla f_{P_n}^\beta(\lambda) = \frac{\lambda - F_{\overline{V}_{P_n}^\beta}(\lambda)}{\left\|\lambda - F_{\overline{V}_{P_n}^\beta}(\lambda)\right\|^2} - \frac{\lambda - F_{V_{P_n}}(\lambda)}{\left\|\lambda - F_{V_{P_n}}(\lambda)\right\|^2}.$$

This yields

$$\frac{\lambda - F_{\overline{V}_{P_n}^\beta}(\lambda)}{\left\|\lambda - F_{\overline{V}_{P_n}^\beta}(\lambda)\right\|^2} = \frac{\lambda - F_{V_{P_n}}(\lambda)}{\left\|\lambda - F_{V_{P_n}}(\lambda)\right\|^2}.$$

Taking the norm in both sides, we get $\left\|\lambda - F_{\overline{V}_{P_n}^\beta}(\lambda)\right\| = \left\|\lambda - F_{V_{P_n}}(\lambda)\right\|$, and finally, replugging this in the equality above, we directly get $F_{\overline{V}_{P_n}^\beta}(\lambda) = F_{V_{P_n}}(\lambda)$, i.e. $\nabla r_{P_n}^\beta(\lambda) = 0$. Let us now prove the second result. We denote $D_n^\beta = \lambda - F_{\overline{V}_{P_n}^\beta}(\lambda)$ and $D_n = \lambda - F_{V_{P_n}}(\lambda)$. We assume now that $\left\|f_{P_n}^\beta(\lambda)\right\| \le \varepsilon$. This inequality gives us two inequalities

$$\left|\frac{1}{\|D_n\|} - \frac{1}{\left\|D_n^\beta\right\|}\right| \le \varepsilon \text{ and } \left\|\frac{D_n}{\|D_n\|^2} - \frac{D_n^\beta}{\left\|D_n^\beta\right\|^2}\right\| \le \varepsilon.$$

Hence, we have

$$\begin{aligned}
\left\|\nabla r_{P_n}^\beta\right\| &= \left\|F_{\overline{V}_{P_n}^\beta}(\lambda) - F_{V_{P_n}}(\lambda)\right\| \\
&= \left\|D_n^\beta - D_n\right\| \\
&= \|D_n\|^2 \left\|\frac{D_n}{\|D_n\|^2} - \frac{D_n^\beta}{\|D_n\|^2}\right\| \\
&\le \|D_n\|^2 \left(\left\|\frac{D_n}{\|D_n\|^2} - \frac{D_n^\beta}{\left\|D_n^\beta\right\|^2}\right\| + \left\|D_n^\beta\right\| \left|\frac{1}{\|D_n\|^2} - \frac{1}{\left\|D_n^\beta\right\|^2}\right|\right) \\
&\le 9B_W^2 B_\Phi^2 \varepsilon + 9B_W^2 B_\Phi^2 \left(1 + \frac{\|D_n\|}{\left\|D_n^\beta\right\|}\right) \left|\frac{1}{\|D_n\|} - \frac{1}{\left\|D_n^\beta\right\|}\right| \\
&\le 27 B_W^2 B_\Phi^2 \varepsilon.
\end{aligned}$$

Let $\lambda \in \mathbb{R}^m$ such that $\left\|\nabla r_{P_n}^\beta(\lambda)\right\| \le \varepsilon$, i.e.

$$\left\|F_{\overline{V}_{P_n}^\beta}(\lambda) - F_{V_{P_n}}(\lambda)\right\| \le \varepsilon. \tag{45}$$

We have

$$\left\| \tilde{\lambda} - F_{\overline{V}^{\beta}_{P_n}}(\tilde{\lambda}) \right\| \leq \left\| \tilde{\lambda} - F_{\overline{V}^{\beta}_{P_n}}(\lambda) \right\| \tag{46}$$

$$\leq \left\| \tilde{\lambda} - F_{V_{P_n}}(\lambda) \right\| + \left\| F_{V_{P_n}}(\lambda) - F_{\overline{V}^{\beta}_{P_n}}(\lambda) \right\| \tag{47}$$

$$\leq \left\| \tilde{\lambda} - F_{V_{P_n}}(\tilde{\lambda}) \right\| + \varepsilon. \tag{48}$$

The last inequality is due to inequality 45 and $F_{V_{P_n}}(\tilde{\lambda}) = F_{\overline{V}^{\beta}_{P_n}}(\lambda)$. We finish by proving this last equality. We have for every $v \in -V_{P_n}$,

$$\left\| \tilde{\lambda} - v \right\|^2 = \left\| \lambda + r \frac{\lambda - F_{V_{P_n}}(\lambda)}{\| \lambda - F_{V_{P_n}}(\lambda) \|} - v \right\|^2 \tag{49}$$

$$= \| \lambda - v \|^2 + r^2 + 2r \left\langle \lambda - v, \frac{\lambda - F_{V_{P_n}}(\lambda)}{\| \lambda - F_{V_{P_n}}(\lambda) \|} \right\rangle \tag{50}$$

$$\geq \| \lambda - v \|^2 + r^2. \tag{51}$$

Notice that when the left-hand side in 51 is minimized (i.e. $v = F_{V_{P_n}}(\lambda)$, the right-hand side takes the same value. Hence, $v = F_{V_{P_n}}(\lambda)$ is also a minimizer of the left-hand side, i.e. $F_{V_{P_n}}(\tilde{\lambda}) = F_{\overline{V}^{\beta}_{P_n}}(\lambda)$. Finally, inequality 48 yields

$$\left\| \tilde{\lambda} - F_{\overline{V}^{\beta}_{P_n}}(\tilde{\lambda}) \right\| - \left\| \tilde{\lambda} - F_{V_{P_n}}(\tilde{\lambda}) \right\| \leq \varepsilon,$$

Which gives

$$r^{\beta}_{P_n}(\tilde{\lambda}) = \frac{1}{2} \left\| \tilde{\lambda} - F_{\overline{V}^{\beta}_{P_n}}(\tilde{\lambda}) \right\|^2 - \frac{1}{2} \left\| \tilde{\lambda} - F_{V_{P_n}}(\tilde{\lambda}) \right\|^2$$

$$= \frac{1}{2} \left( \left\| \tilde{\lambda} - F_{\overline{V}^{\beta}_{P_n}}(\tilde{\lambda}) \right\| - \left\| \tilde{\lambda} - F_{V_{P_n}}(\tilde{\lambda}) \right\| \right) \left( \left\| \tilde{\lambda} - F_{\overline{V}^{\beta}_{P_n}}(\tilde{\lambda}) \right\| + \left\| \tilde{\lambda} - F_{V_{P_n}}(\tilde{\lambda}) \right\| \right)$$

$$\leq \frac{1}{2} \left( \left\| \tilde{\lambda} - F_{\overline{V}^{\beta}_{P_n}}(\tilde{\lambda}) \right\| + \left\| \tilde{\lambda} - F_{V_{P_n}}(\tilde{\lambda}) \right\| \right) \varepsilon$$

$$\leq \frac{1}{2}(6B_W B_\Phi + 2r)\varepsilon \leq 11 B_W B_\Phi \varepsilon.$$

Using this, we also prove the bound on the gradient of $r^{\beta}_{P_n}$. Using the properties of the projection $F_{V_{P_n}}(\tilde{\lambda})$, we have

$$\left\| \nabla r^{\beta}_{P_n}(\tilde{\lambda}) \right\| = \sqrt{ \left\| F_{\overline{V}^{\beta}_{P_n}}(\tilde{\lambda}) - F_{V_{P_n}}(\tilde{\lambda}) \right\|^2 }$$

$$= \sqrt{ \left\| F_{\overline{V}^{\beta}_{P_n}}(\tilde{\lambda}) - \tilde{\lambda} \right\|^2 - \left\| F_{V_{P_n}}(\tilde{\lambda}) - \tilde{\lambda} \right\|^2 - 2 \underbrace{\left\langle F_{\overline{V}^{\beta}_{P_n}}(\tilde{\lambda}) - F_{V_{P_n}}(\tilde{\lambda}), F_{V_{P_n}}(\tilde{\lambda}) - \tilde{\lambda} \right\rangle}_{\geq 0} }$$

$$\leq \sqrt{ r^{\beta}_{P_n}(\tilde{\lambda}) } \leq \sqrt{ 11 B_W B_\Phi \varepsilon }.$$

We finish by proving the bound on $\overline{\theta}^{\beta}_{P_n}(\tilde{\lambda})$. Using Lemma 2, we have $\overline{\theta}^{\beta}_{P_n}(\tilde{\lambda}) = \tilde{\lambda} + F_{\overline{V}^{\beta}_{P_n}}(\tilde{\lambda})$, hence,

$$B_W B_\Phi \leq \left\| \overline{\theta}^{\beta}_{P_n}(\tilde{\lambda}) \right\| \leq 11 B_W B_\Phi.$$

$\square$

## A.17. Proof of Proposition 3

**Proof.** Let $\{\lambda_i\}$ be the sequence of iterates generated by applying gradient descent to $f^{\beta}_{P_n}$. Suppose a subsequence $\{\lambda_{i_t}\}$ of this sequence converges to a limit point $\lambda^\star$. If $\lambda^\star$ is not on the boundary of $-V_{P_n}$ or $-\overline{V}^{\beta}_{P_n}$, then by (Nocedal and Wright, 1999), $\lambda^\star$ is a stationary point of $f^{\beta}_{P_n}$. We now show that if the limit point $\lambda^\star$ lies on the boundary of $-V_{P_n}$, then it must also lie on the boundary of $-\overline{V}^{\beta}_{P_n}$. Indeed, on the one hand, the objective function $f^{\beta}_{P_n}(\lambda_{i_t})$ is non-increasing, so $f^{\beta}_{P_n}(\lambda_{i_t}) \leq f^{\beta}_{P_n}(\lambda_0)$ for all $t$. On the other hand, $-\log(d(\lambda^{i_t}, -V_{P_n})) \to \infty$, which implies that $-\log(d(\lambda^{i_t}, -\overline{V}^{\beta}_{P_n})) \to \infty$. Therefore, $d(\lambda^\star, -\overline{V}^{\beta}_{P_n}) \to 0$, yielding the desired result. $\square$

### A.18. Experimental setting

We set $(d, j) = (20, 5)$ and $W$ to be a polyhedron and written as $W = \{w \in \mathbb{R}^d, \; Aw = b, \; 10 \geq w \geq 0\}$ where $A \in \mathbb{R}^{j \times d}$ ($j \leq d$) and $b \in \mathbb{R}^j$. $W$ is closed and bounded, so Assumption 2 holds. In every experiment, we sample $x \sim \mathcal{N}(0, I)$ while all of its coordinates are conditioned to be between 0 and 10. and the coefficients of $A$ from a standard normal Gaussian distribution, and $b$ to be equal to $A\,|w|$, where $w$ has standard normal random coefficients and $|w|$ is the vector whose coordinates are equal to the absolute value of the coordinates of $w$.

To create a framework where we can enforce misspecification and progressively analyze how the performance of different methods varies with the level of misspecification $\gamma_{\mathrm{dec}}(\mathcal{H})$, we need to define a model that allows for easy adjustment of the misspecification level while still yielding meaningful results. In order to do so, we define for $x \in \mathbb{R}^k$, $c(x)$ to be a linear combination of polynomial functions of $x$. This could be written in a compact way as $c(x) = M\phi(x)$ where $M$ is a real valued matrix, and $\phi(x)$ is a vector whose coordinates are polynomial in $x$, defined by

$$\forall x \in \mathbb{R}^k, \; \phi(x) = \begin{pmatrix} x_1 & \cdots & x_k & x_1 x_2 & \cdots & x_1 \ldots x_k \end{pmatrix}^\top = \left( \prod_{i=1}^k x_i^{y_i} \right)_{y \in \{0,1\}^5, y \neq 0}.$$

Furthermore, we define $\hat{c}_\theta(x)$ for $x \in \mathbb{R}^k$ and $\theta \in \mathbb{R}^m$ as $\hat{c}_\theta(x) = M'(\theta)\tilde{\phi}(x)$ where $\tilde{\phi}(x)$ is equal to $\phi(x)$ truncated to its 5 first coordinates, and $M'(\theta)$ is a matrix representation of $\theta$. To increase the level of misspecification, increase the size of $x$ and consequently the gap between $\phi(x)$ and $\tilde{\phi}(x)$ in terms of number of features. To show that this experimental setting indeed corresponds to the theoretical setting studied in this paper, we can see that both the elements of the hypothesis set and the ground truth can be written for all $x \in \mathbb{R}^k$ as $\hat{c}_\theta(x) = \tilde{\Phi}(x)\theta$ and $c(x) = \Phi(x)\theta^\star$, $\theta^\star \in \mathbb{R}^m$, where

$$\tilde{\Phi}(x) = \begin{pmatrix} \tilde{\phi}(x)^\top & & 0 \\ & \ddots & \\ 0 & & \tilde{\phi}(x)^\top \end{pmatrix} \in \mathbb{R}^{20 \times 100}, \quad \Phi(x) = \begin{pmatrix} \phi(x)^\top & & 0 \\ & \ddots & \\ 0 & & \phi(x)^\top \end{pmatrix} \in \mathbb{R}^{20 \times 20s}.$$

Furthermore, for any matrix $M$, we denote $L_r(M)$ the $r$-th row of $M$. In this case, we have $\theta^\star = \begin{pmatrix} L_1(M) & \cdots & L_{20}(M) \end{pmatrix}^\top \in \mathbb{R}^{20s}$ and $\theta = \begin{pmatrix} L_1(M'(\theta)) & \cdots & L_{20}(M'(\theta)) \end{pmatrix}^\top \in \mathbb{R}^{100}$.

## B. Additional Figures

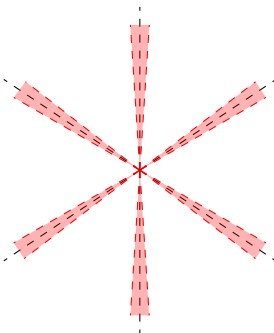

*Figure 4.* Less likely directions for $\hat{c}_\theta(x)$: $\hat{c}_\theta(x)$ is less likely to have a direction included in the red cones, which represent nearly perpendicular directions to one of the faces of the polyhedron (see Assumption 5).

## C. Notation table

| Symbol | Meaning |
| --- | --- |
| $x \in \mathbb{R}^k$ | Context or input features |
| $c \in \mathbb{R}^d$ | True cost vector |

| | |
|---|---|
| $W \subset \mathbb{R}^d$ | Feasible set of decisions (convex and bounded) |
| $w^*(c)$ | Optimal decision under true cost: $\arg\min_{w \in W} c^\top w$ |
| $\Phi(x)$ | Feature transformation (e.g., polynomial basis) applied to context $x$ |
| $\theta \in \mathbb{R}^p$ | Parameters of the predictive model |
| $\hat{c}_\theta(x)$ | Predicted cost vector, typically $\Phi(x)^\top \theta$ |
| $w(\hat{c})$ | Decision minimizing $\hat{c}^\top w$ over $W$ |
| $\mathcal{H}$ | Hypothesis class for cost prediction |
| $P_n$ | Empirical distribution over $n$ i.i.d. samples |
| $\ell_P(\theta)$ | True decision loss under distribution $P$ |
| $\ell_P^\beta(\theta)$ | CILO surrogate loss with cost threshold $\beta$ |
| $g_{P_n}, g_{P_n}^\beta$ | Piecewise linear surrogate components of CILO loss |
| $r_{P_n}^{\beta,\lambda}$ | Smoothed CILO (s-CILO) via Moreau envelope |
| $f_{P_n}^\beta$ | Log-barrier surrogate (log-CILO) |
| $V_{P_n}$ | Set of expected feature-weighted decisions under $P_n$ |
| $\bar{V}_{P_n}^\beta$ | Subset of $V_{P_n}$ with expected cost $\leq \beta$ |
| $\beta$ | Threshold defining near-optimal decision region |
| $\lambda$ | Smoothing parameter used in the Moreau envelope |

# D. Plot with relative regret

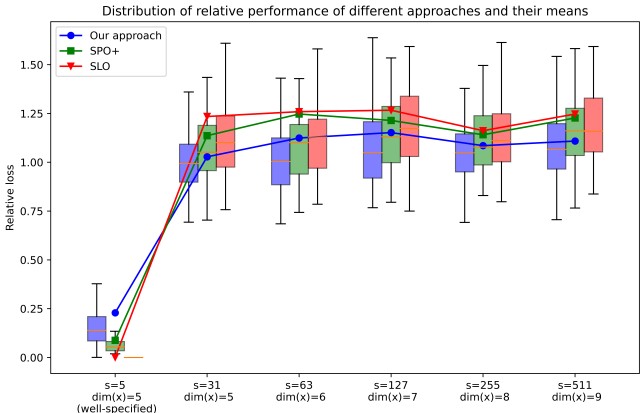

*Figure 5.* Experiment with relative regret results

