# OpenReview forum: "Contextual Optimization Under Model Misspecification: A Tractable and Generalizable Approach"
_ICML.cc/2025/Conference — ICML 2025 poster_

### Official Review · Reviewer_Mq1a · 2025-03-12

**Overall Recommendation:** 4

**Summary:**

This paper presents a new framework for contextual optimization problems where predictive models may not perfectly capture the true underlying relationships. Unlike traditional methods that assume the model is well-specified, this approach introduces a new surrogate loss function designed to ensure that even when predictions are inaccurate, the chosen decisions remain close to optimal. The authors provide theoretical guarantees, including global optimality, generalization performance, and computational efficiency. To handle challenges like non-convexity, they apply smoothing techniques that enable stable gradient-based optimization. Through empirical comparisons, they show that while standard approaches like Sequential Learning and Optimization and Smart Predict-then-Optimize perform well in well-specified cases, their method outperforms them when models are misspecified.

**Claims And Evidence:**

The paper's claims are generally well-supported by theoretical analysis and empirical results. The authors provide formal proofs showing that their surrogate loss function aligns with the true decision objective (Theorem 1) and that minimizing the empirical surrogate loss results in a small out-of-sample decision error (Theorem 2). Additionally, they demonstrate that the method remains tractable for gradient-based optimization through Moreau envelope smoothing (Theorem 4). Overall, the core contributions are justified and well-grounded.

Here are some minor points that could be strengthened:
1. The paper evaluates its method primarily on synthetic data. Expanding the experiments to include real-world datasets from diverse domains would enhance empirical validation. Some theoretical conditions may require verification in practical settings. Providing concrete examples of how these conditions hold in real-world scenarios would strengthen the paper’s contributions.

2. In Theorem 4, global optimality for the proposed surrogate loss is established only for linear hypothesis sets. Extending this result to more general hypothesis classes would improve its applicability.

**Essential References Not Discussed:**

The paper appropriately cites and discusses the most relevant prior works necessary to contextualize its key contributions.

A recent paper that might have some relevance is:
Adam N Elmachtoub, Henry Lam, Haixiang Lan, and Haofeng Zhang. Dissecting the impact of model misspecification in data-driven optimization. In International Conference on Artificial Intelligence and Statistics. PMLR, 2025.

**Experimental Designs Or Analyses:**

I reviewed the validity of the experimental design and analyses. The paper effectively compares its method against Sequential Learning and Optimization (SLO) and Smart Predict-then-Optimize (SPO+), which are the most relevant baselines for decision-focused learning. The evaluation uses synthetic datasets to control for model misspecification. The chosen metrics, particularly decision error, are well-suited to the problem.

**Methods And Evaluation Criteria:**

The proposed methods and evaluation criteria are generally well-aligned with the problem of contextual optimization under model misspecification.

**Other Comments Or Suggestions:**

N/A

**Other Strengths And Weaknesses:**

N/A

**Questions For Authors:**

See Claims And Evidence.

**Relation To Broader Scientific Literature:**

The key contributions of this paper extend existing research in the area of contextual optimization (Sadana et al., 2024). It builds on prior work, such as Smart Predict-then-Optimize (SPO+) (Elmachtoub & Grigas, 2021), which integrates optimization constraints into the learning process to improve decision quality. However, unlike SPO+, which assumes a well-specified prediction model, this paper explicitly accounts for model misspecification. Additionally, the paper connects to research on sequential learning and optimization (SLO) (Donti et al., 2021). The considerations of well-specified and misspecified models also align with findings from Hu et al. (2022) and Elmachtoub et al. (2023), which explore the impact of misspecification on decision-making performance. This work may also relate to advances in robust decision-making under uncertainty, such as distributionally robust optimization (DRO) (Rahimian & Mehrotra, 2019) and end-to-end decision learning (Wilder et al., 2019).

**Theoretical Claims:**

I reviewed the theoretical claims and proofs, such as Theorems 1, 2, 4. Theorem 1 establishes that minimizing the proposed surrogate loss leads to optimal decision policies, and the proof appears mathematically sound. Theorem 2 provides a generalization bound, demonstrating that minimizing empirical loss results in small out-of-sample decision errors, and the proof appears fine. Theorem 4 (and Theorem 5) ensures computational traceability and optimization efficiency, and the proof is well-structured.

---

> ### Author Rebuttal · Authors · 2025-04-01
>
> We thank the reviewer for these thoughtful questions. First, we believe that the paper the reviewer mentioned (Adam N Elmachtoub, Henry Lam, Haixiang Lan, and Haofeng Zhang. Dissecting the impact of model misspecification in data-driven optimization) is indeed relevant to our work, and aim to mention it in our final submission.
>
> Here are the answers for each of the two questions.
>
> 1. This is indeed a valid point. We aim to run more extensive experiments in our future work to validate further the performance of our method, and in particular in real world datasets. We believe that such experiments can strengthen the evidence that shows our approach's empirical performance.
>
> 2. While we focus on a linear hypothesis class $\mathcal{H}$ in the later part of Section~3.3 for clarity and analytical tractability, the core ideas and surrogate loss formulation extend well beyond the linear setting. In particular, suppose $ \Phi : \mathbb{R}^r \times \mathbb{R}^k \rightarrow \mathbb{R}^{d \times m} $ is a smooth, parameterized feature map—for example, a neural network with parameters $u \in \mathbb{R}^r$. We can define a generalized cost predictor as
>
> $$\hat{c}_{u,\theta}(x) = \Phi(x, u)^\top \theta,$$
>
> which retains the structure of a linear combination over learned features while allowing $ \Phi(x, u) $ to be highly expressive.
>
> This setting captures a broad class of nonlinear models such as neural networks where $\theta$ is the weights for the output layer and $u$ is the hidden weight. Under mild regularity conditions (e.g., smoothness of $\Phi$, well-behaved optimization landscapes), the surrogate loss retains the same optimality properties, i.e. every stationary point of our surrogate is a global minimizer. To see why this is true, consider the resulting CILO loss when using this new class of predictors $\ell_{P}^\beta(u,\theta)$ and its Moreau envelope $h_{P}^\beta(\lambda_u,\lambda_\theta)$.  From Theorem 5 page 19, we can say that for any $\lambda_u$ and $\lambda_\theta$, if $u$ and $\theta$ are solutions of the minimization problems resulting from the computation of $h_P^\beta(\lambda_u,\lambda_\beta)$, then if $\frac{\partial h_P^\beta} {\partial \lambda_\theta}(\lambda_u,\lambda_\theta)=0$, then we have $\ell_P^\beta(u,\theta)=0$, i.e. $(u,\theta)$ is a global minimizer of $\ell_P^\beta$. Hence, if $(\lambda_u,\lambda_\theta)$ is a stationary point of $h_P^\beta$, then we have $\frac{\partial h_P^\beta}{\partial \lambda_\theta}(\lambda_u,\lambda_\theta)=0$ and consequently $(u,\theta)$ is a global minimizer of $\ell_P^\beta$.

---

### Official Review · Reviewer_e7We · 2025-03-14

**Overall Recommendation:** 4

**Summary:**

The paper addresses the case of hypothesis class misspecification and proposes a new contextual optimization framework that ensures both tractability (via regularizing) and generalizability.

post-rebuttal: I thank the authors for their response and explanation on generalizability of their ideas. I am keeping my score.

**Claims And Evidence:**

Claims made in the submission are supported by clear and convincing evidence

**Essential References Not Discussed:**

I am not aware of such cases.

**Experimental Designs Or Analyses:**

I checked the soundness of the experimental setup and they stand valid.

**Methods And Evaluation Criteria:**

The methods and evaluation criteria make sense.

**Other Comments Or Suggestions:**

N/A

**Other Strengths And Weaknesses:**

I think the method is original to the field of data-driven decision-making community.

**Questions For Authors:**

How would the analysis of global optimality generalize to non-linear hypothesis classes?

**Relation To Broader Scientific Literature:**

It complements the prior predict-then-optimize literature e.g. Elmachtoub, A. N. and Grigas, P. (2022). Smart “predict, then optimize”. Management Science, 68(1):9–26. and Elmachtoub, A. N., Lam, H., Zhang, H., and Zhao, Y. (2023). Estimate-then-optimize versus integrated-estimationoptimization: A stochastic dominance perspective. arXiv preprint arXiv:2304.06833. by considering specifically model misspecification.

**Theoretical Claims:**

I haven't checked the proofs; though they look highly plausible.

---

> ### Author Rebuttal · Authors · 2025-04-01
>
> We thank the reviewer for this thoughtful question. While we focus on a linear hypothesis class $\mathcal{H}$ in the later part of Section~3.3 for clarity and analytical tractability, the core ideas and surrogate loss formulation extend well beyond the linear setting.
>
> In particular, suppose $ \Phi : \mathbb{R}^r \times \mathbb{R}^k \rightarrow \mathbb{R}^{d \times m} $ is a smooth, parameterized feature map—for example, a neural network with parameters $u \in \mathbb{R}^r$. We can define a generalized cost predictor as
>
> $$\hat{c}_{u,\theta}(x) = \Phi(x, u)^\top \theta,$$
>
> which retains the structure of a linear combination over learned features while allowing $ \Phi(x, u) $ to be highly expressive.
>
> This setting captures a broad class of nonlinear models such as neural networks where $\theta$ is the weights for the output layer and $u$ is the hidden weight. Under mild regularity conditions (e.g., smoothness of $\Phi$, well-behaved optimization landscapes), the surrogate loss retains the same optimality properties, i.e. every stationary point of our surrogate is a global minimizer. To see why this is true, consider the resulting CILO loss when using this new class of predictors $\ell_{P}^\beta(u,\theta)$ and its Moreau envelope $h_{P}^\beta(\lambda_u,\lambda_\theta)$.  From Theorem 5 page 19, we can say that for any $\lambda_u$ and $\lambda_\theta$, if $u$ and $\theta$ are solutions of the minimization problems resulting from the computation of $h_P^\beta(\lambda_u,\lambda_\beta)$, then if $\frac{\partial h_P^\beta} {\partial \lambda_\theta}(\lambda_u,\lambda_\theta)=0$, then we have $\ell_P^\beta(u,\theta)=0$, i.e. $(u,\theta)$ is a global minimizer of $\ell_P^\beta$. Hence, if $(\lambda_u,\lambda_\theta)$ is a stationary point of $h_P^\beta$, then we have $\frac{\partial h_P^\beta}{\partial \lambda_\theta}(\lambda_u,\lambda_\theta)=0$ and consequently $(u,\theta)$ is a global minimizer of $\ell_P^\beta$.

---

### Official Review · Reviewer_modT · 2025-03-16

**Overall Recommendation:** 1

**Summary:**

This paper considers misspecification in the contextual optimization problem, or the predict-then-optimize problem. The authors use a toy example to illustrate the failure of some existing approaches, such as SPO+ and SLO, when the hypothesis class for the prediction part is misspecified. Then, to address this issue, this paper proposes a new traceable surrogate loss function to learn the predictor, and shows the performance of this new method with both theoretical guarantee and numerical performance.

**Claims And Evidence:**

Yes. The authors use a counter-example to show the weakness of some existing methods and also provide proof to show the strength of their new approach.

**Essential References Not Discussed:**

I would not say the work is essential for the predict-then-optimize or the contextual optimization problem. However, potentially [1] can still also address the misspecification issues for the given example. More specifically, it seems that the issues in the toy example come from the misalignment between prediction accuracy in parameter prediction and the accuracy in optimality prediction. Consequently, it is possible that a KKT-based method can address this misspecification issue.

If my statement is corrected, could the authors illustrate more on the necessity of their new approach? I am happy to change my rating based on the authors' answer to this one.

[1] Maximum Optimality Margin: A Unified Approach for Contextual Linear Programming and Inverse Linear Programming

**Experimental Designs Or Analyses:**

I roughly went through the numerical part.

**Methods And Evaluation Criteria:**

Yes. Hypothesis class misspecification is an important setup in learning with optimization.

**Other Comments Or Suggestions:**

It would be better to include more comparisons between the new approach and other methods for the PTO or contextual optimization problem.

**Other Strengths And Weaknesses:**

The strengths have been discussed in the summary. When talking about weaknesses, I wonder whether the authors could elaborate more on the necessity of this new approach. More details can be found in the Essential References part. This is also my major concern.

**Questions For Authors:**

Please see the Essential References part.

**Relation To Broader Scientific Literature:**

This work considers the misspecification issues for the PTO or contextual optimization problem, while to my best knowledge, the existing work needs to assume a well-specified hypotheis class.

**Theoretical Claims:**

I checked the proof for the toy example, and roughly checked the proof for the main theorem.

---

> ### Author Rebuttal · Authors · 2025-04-01
>
> We thank the reviewer bringing up the work of Sun et al. [2023], which proposes a novel approach for learning a cost predictor by maximizing the optimality margin—ensuring that the reduced cost of the predicted solution is positive in the ground-truth optimal basis.
>
> This method can lead to robust decisions under certain conditions. However, we would like to highlight a key assumption made in their analysis (see page 7 of their paper), which implies the existence of a cost predictor in the hypothesis class that yields the same decision as the ground-truth cost. This is equivalent to assuming the hypothesis class is decision well-specified, which we formally define in our paper (Definition 5, page 11). Our framework is explicitly designed to relax this assumption; thus, we address a more general (and practically relevant) setting where no cost predictor in the hypothesis class can yield optimal decisions. To the best of our knowledge, this is the first approach that can minimize the decision cost in a tractable manner under misspecification.
>
> Note that the optimality margin in Sun et al. [2023] focuses on the magnitude of optimality violations (e.g., positivity of reduced costs)– this is still an inconsistent metric with decision cost, since decision quality is ultimately determined by whether the optimal decision under the predicted cost function aligns with that of the true cost.
>
> In contrast, our method directly minimizes the decision error of the optimal decision-making policy under the predicted cost, and we provide guarantees that hold even without decision well-specification (see Theorem 1, page 4).
>
> Although our first toy example (Example 1, page 2) assumes decision well-specification, we clarify after Example 2 (line 170, page 4) that our method continues to perform well even when this assumption fails, while other methods, including SPO+, SLO, and by extension Sun et al., can fail.
> To further support this point, we are including an updated version of Example 2 where the method in Sun et al. [2023] does not yield the optimal cost predictor, while ours does. The core intuition of our example is that a predictor that classifies nearly all points correctly but has extreme optimality constraint violation for one point will not be favored by their approach over a predictor that makes poor decisions consistently but mildly violates the optimality margin, whereas the optimal predictor in terms of decision performance is the one that classifies correctly the most amount of points. Our approach provides the optimal classifier in this setting as well.
>
> Regarding the mention of “KKT-based” methods: if the reviewer was referring to the use of KKT conditions to characterize predictive performance (as in Sun et al. [2023]), we believe the limitations noted above apply. If a different method was intended, we would be happy to provide further clarification upon request.
>
> Consider a refinement of example 2 where we expand the support of the distribution of the context to $\\{1,2,3\\}$. We consider the two cost predictors $\hat{c}_1$ and $\hat{c}_2$ to satisfy $\hat{c}_1(1)=\frac{1}{8}, \hat{c}_1(2)=\frac{1}{8},\hat{c}_1(3)=-100000,\hat{c}_2(1)=1,\hat{c}_2(2)=-\frac{1}{6},\hat{c}_2(3)=-1.$
>
> Recall that in example 1, the ground truth cost is always equal to $1$. The problem we solve to make a decision given a prediction $\hat{c}$ is the following $\max_{w\in[-1/2,1/2]} \hat{c}w.$ In order to apply the approach in Sun et al. [2023], we write the maximization problem above in its standard form.
> \\begin{align*}
> \min_{w_+,w_-,s_u,s_\ell\geq 0}&\\;\begin{pmatrix}
> -\hat{c} \\\\ \hat{c} \\\\ 0 \\\\ 0
> \end{pmatrix}^\top \cdot\begin{pmatrix}
> w_+ \\\\ w_- \\\\ s_u \\\\ s_\ell
> \end{pmatrix}\\\\
> \text{s.t. }&\begin{pmatrix}
> 1 & -1 & 1 & 0 \\\\ -1 & 1 & 0 & 1
> \end{pmatrix}\begin{pmatrix}
>  w_+ \\\\ w_- \\\\ s_u \\\\ s_\ell
> \end{pmatrix}=\begin{pmatrix}
> 1/2 \\\\ 1/2
> \end{pmatrix}.
> \\end{align*}
> Recall that in Sun et al., the optimization problem they solve in order to solve the optimal predictor is in equation 3 page 4 of their paper. We drop the term $\frac{\lambda}{2}\|\|\Theta\|\|_2^2$, although it is possible to keep it and construct a model that gives the same result while keeping this term. This optimization problem in our setting becomes the following
>
> \\begin{align*}
> \min_{\hat{c}\in \\{\hat{c}_1,\hat{c}_2\\}}& \frac{1}{3}(||v_1||_1+||v_2||_1+||v_3||_1)\\\\
> \text{s.t.} &\\; \forall t\in \\{1,2,3\\}, \begin{pmatrix}
> 0 \\\\ \hat{c}(i)
> \end{pmatrix}\geq \begin{pmatrix}
> 1 \\\\ 1
> \end{pmatrix} -v_i
> \\end{align*}
>
> The value of the minimum above for $\hat{c}_1$ is equal to $\sim 33334.66$ and for $\hat{c}_2$ is equal to $\sim 2$, which means that the approach in Sun et al. favors $\hat{c}_2$ even though it is suboptimal, whereas our method favors $\hat{c}_1$.
>
> Finally, we aim to include further comparison with other approaches in our full submission, and are open to compare with any further approaches the reviewer has in mind.

---

> > ### Comment · Reviewer_modT · 2025-04-05
> >
> > I appreciate the authors for the responses, I wonder whether the authors could elaborate more on why the new method favors \hat{c}_1 in the new setting if beta=-1/2.
> >
> > Additionally, although the authors' new example with extreme values is helpful to understand the flaw of [1], but it actually also raises me further concerns of this new approach regarding extreme cases.
> >
> > Specifically, the new example uses extreme values to show the issue of a simplified version of some kkt-based method. However, it seems that the new approach in this paper also suffers from extreme values. If the hypothesis class contains models with extreme small values, such as { \hat{c}_k: \hat{c}_k~1/k, k=1,2,...}\subset H, the learned model from minimizing l^\beta_P might not be very useful. To be more specific, in this new example, if one also has \hat{c}_3(1) = -1/10000000, \hat{c}_3(1) = -1/10000000, \hat{c}_3(1) = -1/10000000, l^\beta_P might prefer \hat{c}_3, which performs worse than \hat{c}_1 and \hat{c}_2.
> >
> > Based on this, I would keep my previous rating.

---

> > > ### Author Response · Authors · 2025-04-05
> > >
> > > Thank you for your reply. We respectfully point out there is some misunderstanding, and we should emphasize that our surrogate provably always outputs the optimal cost predictor in terms of decision performance if we choose suitable $\beta$, which can be obtained by line search.
> > > Please refer to our consistency theorem (Theorem 1 page 4), which guarantees that our surrogate always favors the optimal nonzero cost predictor with smallest decision error.
> > > In our method, we should not always take $\beta=-1/2$ and taking $-1/2$ is incorect in the example.
> > > We should take  $\beta=\beta^\star_{\mathcal H,P}$ which is equal to the minimal possible value of the average decision performance when choosing a predictor from the hypothesis set.
> > >
> > > Hence, in the example we have provided, we should use  $\beta=\beta^\star_{\mathcal H,P}=(-1/2+(-1/2)+1/2)/3=-1/6$, and our surrogate will indeed prioritize $\hat{c}_1$ over $\hat{c}_2$ using $\beta=-1/6$.
> > >
> > > Similarly, when adding $ \hat{c}\_{3} $ satisfying $\hat{c}\_{3} (i) = -1/10000000$  for any $i\in\\{1,2,3\\}$, we still have  $\beta=\beta^\star_{\mathcal H,P}=-1/6$ and our surrogate still favors $\hat{c}_1$. In particular, the surrogate loss values for each predictor is as follows:
> > >
> > > $\ell_P^\beta(\hat{c}_1)=0$, $\ell_P^\beta(\hat{c}_2)=\frac{1}{18}$ and $\ell_P^\beta(\hat{c}_3)=\frac{1}{30000000}$
> > >
> > > If you have further questions or concerns, we are ready to make things more clear.

---

### Official Review · Reviewer_WSCt · 2025-03-22

**Overall Recommendation:** 3

**Summary:**

The paper proposes a new optimization surrogate for contextual linear optimization which is a hard problem due to the nonconvexity of the loss function. The newly proposed surrogate is also non convex but is a difference of convex functions. They prove generalization bounds for their surrogate relative to the original/target loss. While the surrogate has good generalization bounds, it potentially does not converge to a stationary point. Thus, the authors propose applying a Moreau envelope smoothing technique to the surrogate. The smoothed surrogate is then shown to have no "bad" first-order stationary points or local minima. The paper also provide strategies for avoiding zero solutions. Finally, the paper concludes with a shortest path experiment and compares their approach against SPO+ and sequential learning and optimization approaches.

**Claims And Evidence:**

Some of the claims in the paper are hard to verify like Proposition 2.1. In the proof, the authors seem to convert the minimization problems into maximization problems. They seem to claim that $\min c^{\top}x$ is $-\max c^{\top}x$ which seems incorrect. My belief is that the authors used a CILO form for maximization problems but mixed up their steps with the version of the proof using minimization problems.

The authors also claim that existing works lack generalization bounds as they are empirical, however, [1] provides the same type of generalization bounds as this paper and only does not provide global optimality results.

The paper also seems to claim that most surrogates do not consider the misspecified setting. However, the main surrogate that requires well-specified hypothesis classes is just SPO+. Methods like [2] directly optimize SPO loss which would practically also cover misspecified settings.

___
[1] Huang, Michael, and Vishal Gupta. "Decision-focused learning with directional gradients." Advances in Neural Information Processing Systems 37 (2024): 79194-79220.
[2] Jeong, Jihwan, et al. "An exact symbolic reduction of linear smart predict+ optimize to mixed integer linear programming." International Conference on Machine Learning. PMLR, 2022.

**Essential References Not Discussed:**

The paper doesn't seem to mention PyEPO [1] which provide standard numeric benchmarks and methods to evaluate. Many existing surrogates in the PyEPO seem to argue that their surrogates enjoy good landscape. With global optimality guarantees, this paper should be able to verify if such claims are true for existing benchmarks and give insight about what settings are "easy" or "hard" to solve to global optimality for existing surrogates.
____
[1] Tang, Bo, and Elias Boutros Khalil. "Pyepo: A pytorch-based end-to-end predict-then-optimize library with linear objective function." OPT 2022: Optimization for Machine Learning (NeurIPS 2022 Workshop). 2022.

**Experimental Designs Or Analyses:**

The design of the main numerical experiment seems valid, but lacks many details such as the number of samples generated or a main body description of the optimization problem solved. It is also limited due to the lack of problem settings as well as lack of benchmark methods. The PyEPO package [1] is fairly standard benchmark that was not considered by the authors.
____
[1] Tang, Bo, and Elias Boutros Khalil. "Pyepo: A pytorch-based end-to-end predict-then-optimize library with linear objective function." OPT 2022: Optimization for Machine Learning (NeurIPS 2022 Workshop). 2022.

**Methods And Evaluation Criteria:**

The proposed theoretical tools leverage theory for solving non-convex losses which is a key challenge for this class of problems. The numerical evaluation criteria is somewhat limited as there are existing surrogate losses that solve the same problems, however, the paper does not compare against these approaches. This would be helpful for understanding when the proposed surrogate's global optimality guarantees are practically useful. Related, the paper also does not highlight the computational cost of their approach. Optimizing a function with Moreau envelope smoothing does not seem computationally cheap. In contrast existing approaches do not leverage such smoothing and are potentially more computationally efficient. Thus, identifying settings where global optimality is hard to achieve would be made the proposed surrogate approach more practically compelling.

**Other Comments Or Suggestions:**

1) In the proof of Theorem 2, it seems that the CILO loss when lagrangified resembles the PG Loss of [1] with step size $h$ set as the optimal dual variable. It may be worth making the connection.
2) The decision-well-specified definition is hidden in the appendix and not defined in the body even though it is an assumption for Proposition 2.1. It would be helpful to reference it so readers can find it.

**Other Strengths And Weaknesses:**

Strengths
1) The paper constructs a surrogate and shows optimizing the surrogate returns a stationary point that correspond with global optimality. It breaks down the key challenges towards showing such a stationary point can be achieved and addresses each challenge in an organized and clear manner.
2) The paper provides helpful examples for understanding the challenges of misspecification.

Weaknesses
1) The paper lacks details and analysis on the computational components of their approach. First, they do not provide details on how to solve the minimization problems in Definition 2. They also do not highlight the computation cost of their approach in the numerics compared to existing benchmarks.
2) The notation in the paper is confusing. A key definition is $\beta^{\star}\_{\mathcal{H},P} := \min\_{\hat{c}\in\mathcal{H}}\ell\_{P}(\hat{c})$, however, $\ell_P$ seems to only take the input $\theta$. This notation makes it hard to distinguish between $\beta_{\min, P}$ and $\beta^{\star}\_{\mathcal{H},P}$.

**Questions For Authors:**

1) Does an equality similar to the equality between equations (10) and (11) hold for equation (12)?
2) In what settings do $\theta$ converge to 0 if one does not use the log-CILO loss?
3) What properties of the CILO loss allow the Moreau envelope smoothing approach to produce a surrogate with "good" stationary points?
4) It was not clear to me, but is there a way to verify if your choice of $\beta$ returns the global optimal solution for the empirical loss? Another similar question would be, does line search guarantee you obtain the global minimizer for $\theta$ in polynomial time?
5) Does your surrogate practically work for combinatorial problems like shortest path? My main concern is solving the problem with the $\beta$ constraint and the minimization problem in the Moreau envelope smoothing.

**Relation To Broader Scientific Literature:**

This work broadly contributes to the area of decision-focused learning [1]. A key challenge in the literature is constructing computationally tractable surrogates to optimize over as the direct decision-focused loss is non-convex. This paper provides the first results highlighting that there exists surrogates that when optimized over converge to a "good" stationary point.

____
[1] Mandi, Jayanta, et al. "Decision-focused learning: Foundations, state of the art, benchmark and future opportunities." Journal of Artificial Intelligence Research 80 (2024): 1623-1701.

**Theoretical Claims:**

As discussed above, Proposition 2.1 proof seems incorrect. Restated: "In the proof, the authors seem to convert the minimization problems into maximization problems. They seem to claim that $\min c^{\top}x$ is $-\max c^{\top}x$ which seems incorrect. My belief is that the authors used a CILO form for maximization problems but mixed up their steps with the version of the proof using minimization problems. "

I checked the theoretical results up Propostion 1, which seemed correct.

---

> ### Author Rebuttal · Authors · 2025-04-01
>
> Replies to initial remarks: we thank the reviewer for their remarks.
>
> - About Proposition 2.1: In binary classification, we aim to maximize the cost. Hence, if we make a cost prediction $\hat{c}$, then we solve  $\min -\hat{c}^\top w $ to make a decision. Consequently, in the minimization setting, we are making the prediction $-\hat{c}$ and the CILO loss can be written as the same as defined but replacing $\hat{c}$ by $-\hat{c}$. Hence, we believe our proof is correct. A more detailed proof of line 623 page 12 is available here: https://imgur.com/a/NBVRRp4
>
> - About the approach in Jeong et al.: We appreciate the reviewer’s insight. Jeong et al. formulate the SPO loss minimization as an MILP for linear hypothesis sets, whereas our approach is computationally tractable and scalable. It avoids MILP-based optimization, which can be impractical in high dimensions or when fast gradient-based training is needed (e.g., with deep models or large datasets). Our surrogate is differentiable, smoothly approximated via the Moreau envelope, and compatible with standard first-order methods. Our key point is that our approach is the first to tractably minimize decision cost under misspecification with theoretical guarantees.
>
> - Additional details about the experiments: each experiment was conducted using a training dataset of 20 samples and a testing dataset of 20 samples. The optimization was performed using GDA/gradient descent with a constant step size applied to the least squares loss, SPO+, and our smoothed CILO loss. We will make sure to highlight this more clearly in the final submission.
>
> - We agree that broader benchmarks would enhance the practical relevance of our method. Our minimalistic, controlled design follows Hu et al. (https://arxiv.org/abs/2011.03030), enabling systematic analysis of decision quality under misspecification and direct links to theory. While we did not include PyEPO methods, we see PyEPO as a valuable benchmark and appreciate the suggestion. Expanding comparisons to PyEPO and larger benchmarks is a promising future direction once our core theoretical contributions are validated.
>
> - While our goal is not optimal complexity, our primary aim is demonstrating that contextual optimization under misspecification is tractable using a surrogate loss optimized with first-order methods. For clarity, we use Gradient Descent-Ascent (GDA-max) in our experiments, though advanced min-max algorithms could improve the complexity. Our surrogate’s Moreau envelope minimization can be reformulated as a min-max problem (https://imgur.com/a/fYxXQAk), enabling efficient optimization via the single-loop smoothed gradient descent-ascent algorithm (https://arxiv.org/abs/2010.15768). Since $w_P(x_1),\dots,w_P(x_n)$ have independent constraints, parallelization is possible. While we prioritized conceptual clarity, integrating accelerated methods is a promising direction for future work, which we will clarify and support with relevant citations.
>
> - About the PG loss in Huang et al.: The approach in Huang et al. can be seen as the penalty method to solve the bilevel optimization problem $\min_{\hat{c}^\top w\leq \min_{w'\in W}\hat{c}^\top w'}c^\top w$. In our approach, we flip the upper and lower level, which is why our loss when langragified appears similar (proof: https://imgur.com/a/h0KbcjC). Our approach has global optimiality guarantees, and also differs by the fact that the lagrange multiplier in our surrogate is bounded (see Appendix A.9) and the "lagrange multiplier" $\frac{1}{h}$ in the PG loss has to be large enough when the sample size is large which can cause numerical issues.
>
> Replies to the questions:
> 1. The realizations of $w_P(x)$ in equaltions (10) and (11) are independently constrained, whereas in equation (12), the realizations of $w_P^\beta(x)$ are linked by the constraint $\mathbb E(c^\top w_P^\beta (x))\leq \beta$ and hence the same inequality does not hold.
>
> 2. It seems that in most practical settings, this phenomenon does not happen. However, since we do not have a theoretical characterization of when this happens, we introduced the log-CILO loss, which helps to address this issue.
>
> 3. In reality, the good landscape properties of the CILO loss remain even before considering its Moreau envelope. Indeed, in Appendix A.13, a more complete version of theorem 4 is provided, where we show that any stationary point of the non-smooth version of the CILO loss is a global minimizer. The Moreau envelope is used only as a tool to tractably minimize the CILO loss.
>
> 4. Theoretically, we can prove that line search with precision $\epsilon$ provides an global minimizer with precision $\epsilon$. Proof:https://imgur.com/a/LHxvfkk
>
> 5. The shortest path problems are within our problem setting--their decision-making problems are linear programs. So we can directly apply our approach to solve them, but we agree that testing our algorithms in practical settings can be interesting and important future directions.

---

> > ### Comment · Reviewer_WSCt · 2025-04-03
> >
> > Thank you for your responses, they really helped clarify some ambiguity in the text!
> >
> > 1. Proposition 2.1: Thank you for the additional clarification, in the text it was a bit ambiguous what specific problem you were trying to solve and your additional note was helpful.
> > 2. Connections with PG Loss: Thank you for your insight about the relationship between CILO and PG Loss. It’s interesting see the connection through the bilevel optimization lens.
> >
> > ## Additional Questions About Computational tractability
> >
> > I have some additional questions related to the computational tractability of your approach. It seems to be a large selling point of the paper, so I wanted to double check a few things.
> > 1. Computational tractability: You mention
> > >The shortest path problems are within our problem setting--their decision-making problems are linear programs. So we can directly apply our approach to solve them, ...
> >
> >  If $W_P$ maps into a discrete space, doesn’t adding the $\beta$ constraint for $W_P^{\beta}$ potentially change a potentially tractable combinatorial problem into a computationally challenging integer program? For example, shortest path can be solved as a LP because the constraints are totally unimodular. However, adding the $\beta$ constraint would remove that property and make it a hard to solve integer program. Is there something I’m missing that makes the auxiliary problem you solve tractable? If not it might be worth highlighting that CILO is only tractable for linear programs, but not necessarily combinatorial problems.
> >
> > 2. Line Search result: Can you provide some more details for the line search proof? I am a little confused how you showed it’s $O_p(\epsilon)$ optimal where $\epsilon$ is the precision. It sounds like line search finds the minimizing beta for $\ell_P(\theta(\beta))$ where $\theta(\beta)$ is the minimizer for $\ell_P^{\beta}$. If $\ell_P(\theta(\beta))$ is not convex in $\beta$ (which intuitively seems true), how does line search not get stuck at a local optimum?
> >
> > ## Response to Response
> > Writing this here since I can't add an additional comment.
> > 1. To clarify I am talking about the problem $\min_{w_P^{\beta} \in W_P^{\beta}} \mathbb{E} [ \hat{c}_{\theta}(x)^{\top} w_P^{\beta}(x)]$ and equation (12) being intractable. How do you solve this problem practically? You can't use a linear relaxation because you have the additional constraint $E[ c^{\top} w_P(x) ] \le \beta$ to deal with.
> >
> > If you are suggesting you are searching over the space of mappings, it is unclear to me how you are doing so. I could not find a tractable approach in this paper.
> >
> > My assumption was that for the $P_n$ problem you solved $\min \sum_{j=1}^n \hat{c}_{\theta}(x_j)^{\top} w_j$ such that $w_j \in W \forall j$ and $\frac{1}{n} c_j^{\top} w_P(x_j) \le \beta$. You cannot relax the integer constraints in this setting and still expect an integer solution.
> >
> > 2. Maybe using the term "grid search" might be a better alternative to "line search"
> >
> > ## Response to Response 2
> >
> > Thank you for finding this and my previous response!
> >
> > 1. I agree you can always solve a relaxed version of the problem for the surrogate, i.e.,
> > $\min \sum_{j=1}^n \hat{c}_{\theta}(x_j)^{\top} w_j \text{ s.t. } \frac{1}{n} \sum_j c_j^\top w(x_j) \leq \beta, w(x_j)\in W \, \forall j.$
> > However, in such cases, like you mentioned $w_P^{\beta}(x)$ does not need to be an integer variable. As a result, it is very likely your theoretical guarantees about global optimality fail to hold for such combinatorial problems **unless you sacrifice computational tractability**.
> >
> > In your "line search" proof, you used the fact that $\frac{1}{n} \sum_j c_j^\top w_P^{\beta}(x_j) \leq \beta$ holds for a feasible solution to the original problem. But for combinatorial problems, $w_P^{\beta}(x_j)$ may be non-integer and thus infeasible. So if you use the $\hat{c}$ from the surrogate and plug it into the original combinatorial problem, you may get a solution $w_P(x_j)$ with $c_j^\top w_P(x_j) \geq \beta$.
> >
> > Relaxing integer constraints can also cause other issues: i) $\ell_P^{\beta}(\theta)$ might be 0, yet $w_P$ and $w_P^{\beta}$ differ, so the sub-gradient isn’t zero and you’re not at a stationary point.
> > ii) $\ell_P^{\beta}(\theta)$ might be negative, violating Lemma 1.
> > iii) Theorem 1 could be problematic since $\ell_P(\theta) \le \beta$ may not hold even if $\theta$ minimizes $\ell_P^{\beta}(\theta)$.
> >
> > ## Response to Response 3
> > Thank you for humoring my questions! I think what you said plus me working through some proofs makes me believe the issue I raised about combinatorial problems is not a big issue. I've raised my score accordingly.
> >
> > I would recommend if possible:
> > i) Incorporating some of the clearer notation into the proofs.
> > ii) Adding some visualization of the CILO loss landscape.
> > iii) Visualizing the effect of changing $\beta$.
> > iv) More details on implementation.
> >
> > Final question, why can't you use bisection search to find $\beta^{\star}_{\mathcal{H},P}$?

---

> > > ### Author Response · Authors · 2025-04-04
> > >
> > > # EDIT: REPLY 3
> > >
> > > Thank you for your positive feedback! We greatly appreciate your suggestions and will make sure to incorporate them in our full submission. As for your question about bisection search, if you are asking about binary search of $\beta^\star_{\mathcal H,P}$, we so far do not have a clear way to verify whether we have $\beta \geq \beta^\star_{\mathcal H,P}$ using training data.
> > >
> > > Previous replies:
> > >
> > > reply 1: https://hastebin.com/share/unewilakob.swift
> > >
> > > reply 2: https://hastebin.com/share/uyadimipaf.scss
> > >
> > > reply 3: https://hastebin.com/share/awozahovel.ruby

---

### Official Review · Reviewer_cgak · 2025-03-23

**Overall Recommendation:** 3

**Summary:**

In contextual optimization, real-world settings often suffer from model misspecification, meaning the chosen predictor family does not include the true cost function. While existing contextual optimization literature largely focused on well-specified models, this paper tackles that gap by introducing a surrogate loss (“CILO”) that explicitly accounts for model misspecification. The authors show that 1) this surrogate loss is consistent, 2) minimizing the empirical version of this loss has good generalization guarantees, and 3) there exists computationally efficient ways to optimize the proposed surrogate loss. Improvements compared to existing methods are shown in the experiments.

**Claims And Evidence:**

The claims in the paper are supported by evidence.

**Essential References Not Discussed:**

Since a key contribution in sections 3.1 and 3.2 is proving generalization bound under misspecification for contextual optimization under misspecification, the authors should compare to Theorem 2 in "Contextual Linear Optimization with Bandit Feedback" by Hu at al (2024) published in NeurIPS 2024. Theorem 2 therein also considers a generalization bound under misspecification, and they seem to use a similar assumption as Assumption 5 in the current paper.

**Experimental Designs Or Analyses:**

The experiments make sense overall. I have a few questions that I want to authors to clarify:
- Some design choices in the experiments are unclear, e.g., what the phi function is.
- A lot of the existing literature use a normalized loss when reporting the results (e.g., Elmachtoub and Grigas), which makes it clear how much worse the methods are compared to the ground truth. The currect paper reports absolute regret, making it hard to observe the relative improvement.
- The difference between SPO+ and SLO seem to be smaller than I expected in the misspecified setting.

**Methods And Evaluation Criteria:**

The proposed methods and evaluation make sense for the problem.

**Other Comments Or Suggestions:**

N/A

**Other Strengths And Weaknesses:**

I think the contribution of the currect paper is nice. It highlights the under-explored issue of model misspecification in the contextual optimization literature, and the proposed surrogate loss makes sense. I have a few questions that I want to authors to answer, which I listed in the Questions section.

**Questions For Authors:**

I want to briefly summarize my questions in the previous sections:
1. Regarding the theory, please explain what makes optimizing surrogate loss hard if the hypothesis class is non-linear.
2. Regarding the experiments, please (i) specify the phi functions; (ii) consider changing the absolute regret to normalized regret so it's easier to see the relative improvement compared to baseline; (iii) provide an explanation why the performance of SPO+ and SLO are rather similar.
3. Regarding literature, please compare the currect result to Hu et al (2024) "Contextual Linear Optimization with Bandit Feedback". There seems to be some similarities between the generalization bound in the currect paper and Theorem 2 in the referenced paper, and I think the authors should clarify the difference.
4. Regarding presentation, it would be better if the authors were able to make a list of the notations in the appendix, since there are many of them and sometimes it's hard for the reader to keep track of things.

**Relation To Broader Scientific Literature:**

The key contribution of the current paper is to design a computationally-efficient surrogate loss for contextual linear optimization under model misspecification. In comparison, existing literature largely focused on well-specified models.

**Theoretical Claims:**

There's no proof in the main text, and I only very briefly checked some in the appendix. The conclusions in the theoretical claims make intuitive sense. One reservation I have for the theory is that, starting from the later part in section 3.3, the authors assume the hypothesis set H is linear, which is somewhat restrictive.

---

> ### Author Rebuttal · Authors · 2025-04-01
>
> Thank you for there insightful questions. Here are our answers in order.
>
> 1. We thank the reviewer for this thoughtful question. While we focus on a linear hypothesis class $\mathcal{H}$ in the later part of Section~3.3 for clarity and analytical tractability, the core ideas and surrogate loss formulation extend well beyond the linear setting.
>
> In particular, suppose $ \Phi : \mathbb{R}^r \times \mathbb{R}^k \rightarrow \mathbb{R}^{d \times m} $ is a smooth, parameterized feature map—for example, a neural network with parameters $u \in \mathbb{R}^r$. We can define a generalized cost predictor as
>
> $$\hat{c}_{u,\theta}(x) = \Phi(x, u)^\top \theta,$$
>
> which retains the structure of a linear combination over learned features while allowing $ \Phi(x, u) $ to be highly expressive.
>
> This setting captures a broad class of nonlinear models such as neural networks where $\theta$ is the weights for the output layer and $u$ is the hidden weight. Under mild regularity conditions (e.g., smoothness of $\Phi$, well-behaved optimization landscapes), the surrogate loss retains the same optimality properties, i.e. every stationary point of our surrogate is a global minimizer. To see why this is true, consider the resulting CILO loss when using this new class of predictors $\ell_{P}^\beta(u,\theta)$ and its Moreau envelope $h_{P}^\beta(\lambda_u,\lambda_\theta)$.  From Theorem 5 page 19, we can say that for any $\lambda_u$ and $\lambda_\theta$, if $u$ and $\theta$ are solutions of the minimization problems resulting from the computation of $h_P^\beta(\lambda_u,\lambda_\beta)$, then if $\frac{\partial h_P^\beta} {\partial \lambda_\theta}(\lambda_u,\lambda_\theta)=0$, then we have $\ell_P^\beta(u,\theta)=0$, i.e. $(u,\theta)$ is a global minimizer of $\ell_P^\beta$. Hence, if $(\lambda_u,\lambda_\theta)$ is a stationary point of $h_P^\beta$, then we have $\frac{\partial h_P^\beta}{\partial \lambda_\theta}(\lambda_u,\lambda_\theta)=0$ and consequently $(u,\theta)$ is a global minimizer of $\ell_P^\beta$.
>
> 2.(i). More details (including what the function phi is) are available in appendix A.18 in page 24.
>
> 2.(ii). An updated version of our experiment figure with the relative regret is available here: https://imgur.com/a/0rqF2iE
> The boxplots show the distribution of the performance of each method, and the curve shows the mean performance of each method.
>
> 2.(iii) When checking the mean and the median of the absolute difference between the performance of SPO+ and SLO, they range respectively between 9% and 15% and between 7% and 10% even though the median of the difference between the performance of the two methods appears to be small. We believe that such a difference does not suggest that there is some link between SLO and SPO+.
>
> 3.  We thank the reviewer for highlighting the connection to Hu et al. (2024), “Contextual Linear Optimization with Bandit Feedback.” While our assumption is similar in spirit to theirs—both aim to rule out degenerate cases—there is a key conceptual difference.
>
> In Hu et al., the non-degeneracy condition is imposed on the ground-truth cost function, which is unknown and problem-dependent. In contrast, our assumption is placed on the hypothesis class, which is fully under the control of the decision maker. This makes our assumption both practical and easier to verify in real-world applications, since the designer can ensure it holds through appropriate model selection. In Hu et al., by contrast, the assumption may or may not hold depending on the unknown environment.
>
> This distinction also highlights a difference in perspective: our framework is designed to be robust under misspecification, where the true cost function may not lie within the hypothesis class. We will clarify this distinction in the revised manuscript, particularly in the discussion surrounding Theorem~2.
>
> 4. In the revised version, we will include a dedicated table of notations in the appendix summarizing all key symbols and their meanings. This will help improve clarity and ease of reference. Meanwhile, we provide a table of notations in the following link: https://imgur.com/a/73fBZcT

---

### Decision · Program_Chairs · 2025-05-01

**Decision:**

Accept (poster)

**Comment:**

I enjoyed the paper and concur with the overall positive response from the reviewers and happily recommend acceptance. There are a few points, however, that I think are necessary to address in a camera ready. One is an expanded discussion of statistical guarantees in the misspecified setting (e.g. Hu et al 2024 as mentioned by Reviewer cgak) as well as a clearer explanation of the use of a margin condition and its relation to statistical guarantees. Another is a clarification and better discussion of the claimed tractability and its meaning as "practical tractability" vs "theoretical tractability," given that, while well motivated, the reduction is still to a technically "hard" problem in the general setting. These are only what I think are the most crucial points, but the authors should also address all other concerns and certainly incorporate elements from the discussion into the final paper.